# Optimal plasticity for memory maintenance during ongoing synaptic change

**Dhruva V Raman\*, Timothy O'Leary\***

Department of Engineering, University of Cambridge, Cambridge, United Kingdom

**Abstract** Synaptic connections in many brain circuits fluctuate, exhibiting substantial turnover and remodelling over hours to days. Surprisingly, experiments show that most of this flux in connectivity persists in the absence of learning or known plasticity signals. How can neural circuits retain learned information despite a large proportion of ongoing and potentially disruptive synaptic changes? We address this question from first principles by analysing how much compensatory plasticity would be required to optimally counteract ongoing fluctuations, regardless of whether fluctuations are random or systematic. Remarkably, we find that the answer is largely independent of plasticity mechanisms and circuit architectures: compensatory plasticity should be at most equal in magnitude to fluctuations, and often less, in direct agreement with previously unexplained experimental observations. Moreover, our analysis shows that a high proportion of learning-independent synaptic change is consistent with plasticity mechanisms that accurately compute error gradients.

**\*For correspondence:**
dvr23@cam.ac.uk (DVR);
tso24@cam.ac.uk (TO'L)

## Introduction

Learning depends upon systematic changes to the connectivity and strengths of synapses in neural circuits. This has been shown across experimental systems (*Moczulska et al., 2013*; *Lai et al., 2012*; *Hayashi-Takagi et al., 2015*) and is assumed by most theories of learning (*Hebb, 1949*; *Bienenstock et al., 1982*; *Gerstner et al., 1996*).

Neural circuits are required not only to learn, but also to retain previously learned information. One might therefore expect synaptic stability in the absence of an explicit learning signal. However, many recent experiments in multiple brain areas have documented substantial ongoing synaptic modification in the absence of any obvious learning or change in behaviour (*Attardo et al., 2015*; *Pfeiffer et al., 2018*; *Holtmaat et al., 2005*; *Loewenstein et al., 2015*; *Yasumatsu et al., 2008*; *Loewenstein et al., 2011*).

This ongoing synaptic flux is heterogeneous in its magnitude and form. For instance, the expected lifetime of dendritic spines in mouse CA1 hippocampus has been estimated as 1–2 weeks (*Attardo et al., 2015*). Elsewhere in the brain, over 70% of spines in mouse barrel cortex are found to persist for 18 months (*Zuo et al., 2005*), although these persistent spines exhibited large deviations in size over the imaging period (on average, a >25% deviation in spine head diameter).

The sources of these ongoing changes remain unaccounted for, but are hypothesised to fall into systematic changes associated with learning, development and homeostatic maintenance, and unsystematic changes due to random turnover (*Rule et al., 2019*; *Mongillo et al., 2017*; *Ziv and Brenner, 2018*). A number of experimental studies have attempted to disambiguate and quantify the contributions of different biological processes to overall synaptic changes, either by directly interfering with synaptic plasticity, or by correlating changes to circuit-wide measurements of ongoing physiological activity (*Nagaoka et al., 2016*; *Quinn et al., 2019*; *Yasumatsu et al., 2008*; *Minerbi et al., 2009*; *Dvorkin and Ziv, 2016*). Consistently, these studies find that the total rate of

ongoing synaptic change is reduced by only 50% or less in the absence of neural activity or when plasticity pathways are blocked.

Thus, the bulk of steady-state synaptic changes seem to arise from fluctuations that are independent of activity patterns at pre/post synaptic neurons or known plasticity induction pathways. As such, it seems unlikely that their source is some external learning signal or internal reconsolidation mechanism. This is surprising, because maintenance of neural circuit properties and learned behaviour would intuitively require changes across synapses to be highly co-ordinated. To our knowledge, there is no theoretical account or model prediction that explains these observations.

One way of reconciling stable circuit function with unstable synapses is to assume that ongoing synaptic changes are localised to 'unimportant' synapses, which do not affect circuit function. While this may hold in particular circuits and contexts (*Mongillo et al., 2017*), at least some of the ongoing synaptic changes are likely associated with ongoing learning, which must somehow affect overall circuit function to be effective (*Rule et al., 2020*). Furthermore, this model does not account for the dominant contribution of fluctuations among those synapses that do not remain stable over time.

In this work we explore another, non-mutually exclusive hypothesis that active plasticity mechanisms continually maintain the overall function of a neural circuit by compensating changes that degrade memories and learned task performance. This fits within the broad framework of memory maintenance via internal replay and reconsolidation, a widely hypothesised class of mechanisms for which there is widespread evidence (*Carr et al., 2011*; *Foster, 2017*; *Nader and Einarsson, 2010*; *Tronson and Taylor, 2007*).

Compensatory plasticity can be induced by external reinforcement signals (*Kappel et al., 2018*), interactions between different brain areas and circuits (*Acker et al., 2018*), or spontaneous, network-level reactivation events (*Fauth and van Rossum, 2019*). Either way, we can conceptually divide plasticity processes into two types: those that degrade previously learned information, and those that protect against such degradation. We will typically refer to memory-degrading processes as 'fluctuations'. While these may be stochastic in origin, for example due to intrinsic molecular noise in synapses, we do not demand that this is the case. Fluctuations will therefore account for any synaptic change, random or systematic, that disrupts stored information.

The central question we address in this work is how compensatory plasticity should act in order to optimally maintain stored information at the circuit level, in the presence of ongoing synaptic fluctuations. To do this, we develop a general modelling framework and conduct a first-principles mathematical analysis that is independent of specific plasticity mechanism and circuit architectures. We find that the rate of compensatory plasticity should not exceed that of the synaptic fluctuations, in direct agreement with experimental measurements. Moreover, fluctuations should dominate as the precision of compensatory plasticity mechanisms increases, where 'precision' is defined as the quality of approximation of an error gradient. This provides a potential means of accounting for differences in relative magnitudes of fluctuations in different neural circuits. We validate our theoretical predictions through simulation. Together, our results explain a number of consistent but puzzling experimental findings by developing the hypothesis that synaptic plasticity is optimised for dynamic maintenance of learned information.

## Results

### Review of key experimental findings

To motivate the main analysis in this paper we begin with a brief survey of quantitative, experimental measurements of ongoing synaptic dynamics. These studies, summarised in *Table 1*, provide quantifications of the rates of systematic/activity-dependent plasticity relative to ongoing synaptic fluctuations.

We focused on studies that measured 'baseline' synaptic changes that occur outside of any behavioural learning paradigm, and which controlled for stimuli that may induce widespread changes in synaptic strength. The approaches fall into two categories:

1.  Those that chemically suppress neural activity, and/or block known synaptic plasticity pathways, quantifying consequent changes in the rate of synaptic dynamics, in vitro (*Yasumatsu et al., 2008*; *Minerbi et al., 2009*; *Quinn et al., 2019*) and in vivo (*Nagaoka et al., 2016*). The latter study included a challenging experiment in which neural

**Table 1.** Synaptic plasticity rates across experimental models, and the effect of activity suppression.

| Reference | Experimental system | Total baseline synaptic change | % synaptic change that is activity / learning-independent |
|---|---|---|---|
| *Pfeiffer et al., 2018* | Adult mouse hippocampus | 40% turnover over 4 days | NA |
| *Loewenstein et al., 2011* | Adult mouse auditory cortex | >70% of spines changed size by >50% over 20 days | NA |
| *Zuo et al., 2005* | Adult mouse (barrel, primary motor, frontal) cortex | 3–5% turnover over 2 weeks for all regions. 73.9 ± 2.8% of spines stable over 18 months (barrel cortex) | NA |
| *Nagaoka et al., 2016* | Adult mouse visual cortex | 8% turnover per 2 days in visually deprived environment. 15% in visually enriched environment. 7–8% in both environments under pharmacological suppression of spiking. | ≈50% (turnover) |
| *Quinn et al., 2019* | Glutamatergic synapses, dissociated rat hippocampal culture | 28 ± 3.7% of synapses formed over 24 hr period. 28.6 ± 2.3% eliminated. Activity suppression through tetanus neurotoxin -light chain. Plasticity rate unmeasured. | ≈75% (turnover) |
| *Yasumatsu et al., 2008* | CA1 pyramidal neurons, primary culture, rat hippocampus | Measured rates of synaptic turnover and spine-head volume change. Baseline conditions vs activity suppression (NMDAR inhibitors). Turnover rates: 32.8 ± 3.7% generation/elimination per day (control) vs 22.0 ± 3.6% (NMDAR inhibitor). Rate of spine-head volume change: | ≈67 ± 17% (turnover). Size-dependent, but consistently >50% (spine-head volume) |
| *Dvorkin and Ziv, 2016* | Glutamatergic synapses in cultured networks of mouse cortical neurons | Partitioned commonly innervated (CI) synapses sharing same axon and dendrite, and non-CI synapses. Quantified covariance in fluorescence change for CI vs non-CI synapses to estimate relative contribution of activity histories to synaptic remodelling | 62–64% (plasticity) |
| *Minerbi et al., 2009* | Rat cortical neurons in primary culture | Created '*relative synaptic remodeling measure*' (RRM) based on frequency of changes in the rank ordering of synapses by fluorescence. Compared baseline RRM to when neural activity was suppressed by tetrodotoxin (TTX). RRM: 0.4 (control) vs 0.3 (TTX) after 30 hr. | ≈75% (plasticity) |
| *Kasthuri et al., 2015* | Adult mouse neocortex (Three-dimensional *post mortem* reconstruction using electron microscopy). | Data on 124 pairs of 'redundant' synapses sharing a pre/post-synaptic neuron was analysed in *Dvorkin and Ziv, 2016*. They calculated the correlation coefficient of spine volumes and post-synaptic density sizes between redundant pairs. This should be one if pre/post-synaptic activity history perfectly explains these variables. | 77% (post-synaptic density, $r^2 = 0.23$). 66% (spine volume, $r^2 = 0.34$) |
| *Ziv and Brenner, 2018* | Literature review across multiple systems | '*Collectively these findings suggest that the contributions of spontaneous processes and specific activity histories to synaptic remodeling are of similar magnitudes*' | ≈50% |

activity was pharmacologically suppressed in the visual cortex of mice raised in visually enriched conditions.

2. Those that compare 'redundant' synapses sharing pre and post-synaptic neurons, and quantify the proportion of synaptic strength changes attributable to spontaneous processes independent of their shared activity history. These included in vitro studies that involved precise longitudinal imaging of dendritic spines in cultured cortical neurons (*Dvorkin and Ziv, 2016*). They also included in vivo studies, that used electron microscopy to reconstruct and compare the sizes of redundant synapses (*Kasthuri et al., 2015*) *post mortem*.

The studies in *Table 1* consistently report that the the main component (more than 50%) of baseline synaptic dynamics is due to synaptic fluctuations that are independent of neural activity and/or easily identifiable plasticity signals. This is surprising because such a large contribution of fluctuations might be expected to disrupt circuit function. A key question that we address in this study is whether such a large relative magnitude of fluctuations can be accounted for from first principles, assuming that neural circuits need to protect overall function against perturbations.

The hypothesis we assumed is that some active plasticity mechanism compensates for the degradation of a learned memory trace or circuit function caused by ongoing synaptic fluctuations. We will thus express overall plasticity as a combination of *synaptic fluctuations* (task-independent

processes that degrade memory quality) and *compensatory plasticity*, which counteracts this effect. There are various ways such a compensatory mechanism might access information on the integrity of overall circuit function, memory quality or 'task performance'. It could use external reinforcement signals (*Kappel et al., 2018*; *Rule et al., 2020*). Alternatively, such information could come from another brain region, as hypothesised in for example *Acker et al., 2018*, where cortical memories are stabilised by hippocampal replay events. Spontaneous, network-level reactivation events internal to the neural circuit itself could also plausibly induce performance-increasing plasticity (*Fauth and van Rossum, 2019*). Regardless, the decomposition of total ongoing plasticity into fluctuations and systematic plasticity allows us to derive relationships between both that are independent of the underlying mechanisms, which are not the focus of this study.

We must acknowledge that it is difficult, experimentally, to pin down and control for all physiological factors that regulate synaptic changes, or indeed to measure such changes accurately. However, even if one does not take the observations in *Table 1* – or their interpretation – at face value, the conceptual question we ask remains relevant for any neural circuit that needs to retain information in the face of ongoing synaptic change.

## Modelling setup

Suppose a neural circuit is maintaining previously learned information on a task. The circuit is subject to task-independent synaptic fluctuations which can degrade the quality of learned information. Meanwhile, some compensatory plasticity mechanism counteracts this degradation. Throughout this paper, we treat 'memory' and 'task performance' as interchangeable because our framework analyses the effect of synaptic weight change on overall circuit function. In this context, we ask:

*if a network optimally maintains learned task performance, what rate of compensatory plasticity is required relative to the rate of synaptic fluctuations?*

By 'rate' we mean magnitude of change in a given time interval. Our setup is depicted in *Figure 1*. We make the following assumptions, which are also stated mathematically in *Box 1*:

1. The neural network has $N$ adaptive elements that we call 'synaptic weights' for convenience, although they could include parameters controlling intrinsic neural excitability. We represent these elements through a vector $\mathbf{w}(t)$, which we call the neural network state. Changes to $\mathbf{w}(t)$ correspond to plasticity.
2. Any state $\mathbf{w}(t)$ is associated with a quantifiable (scalar) level of task error, denoted $F[\mathbf{w}(t)]$, and called the loss function. A higher value of $F[\mathbf{w}(t)]$ implies greater corruption of previously learned information.
3. The network state can be varied continuously. Task error varies smoothly with respect to changes in $\mathbf{w}(t)$.
4. At any point of time, we can represent the rate of change (i.e. time-derivative) of the synaptic weights as

$$\dot{\mathbf{w}}(t) = \dot{\mathbf{c}}(t) + \dot{\epsilon}(t).$$

as discussed previously, which correspond to compensatory plasticity and synaptic fluctuations, respectively.

The magnitude and direction of plasticity may or may not change continually over time. Correspondingly, we may pick an appropriately small time interval, $\Delta t$, (which is not necessarily infinitesimally small) over which the directions of plasticity can be assumed constant, and write

$$\Delta \mathbf{w}(t) = \Delta \mathbf{c}(t) + \Delta \epsilon(t), \tag{1}$$

where for any time-dependent variable $x(t)$, we use the notation $\Delta x(t) := x(t + \Delta t) - x(t)$. We regard $\Delta \mathbf{c}(t)$ and $\Delta \epsilon(t)$ as coming from unknown probability distributions, which obey the following constraints:

- Synaptic fluctuations $\Delta \epsilon(t)$: We want to capture 'task independent' plasticity mechanisms. As such, we demand that the probability of the mechanism increasing or decreasing any particular synaptic weight over $\Delta t$ is independent of whether such a change increases or decreases task error. A trivial example would be white noise, but systematic mechanisms, such as homeostatic plasticity, could also contribute (*O'Leary, 2018*; *O'Leary and Wyllie, 2011*).

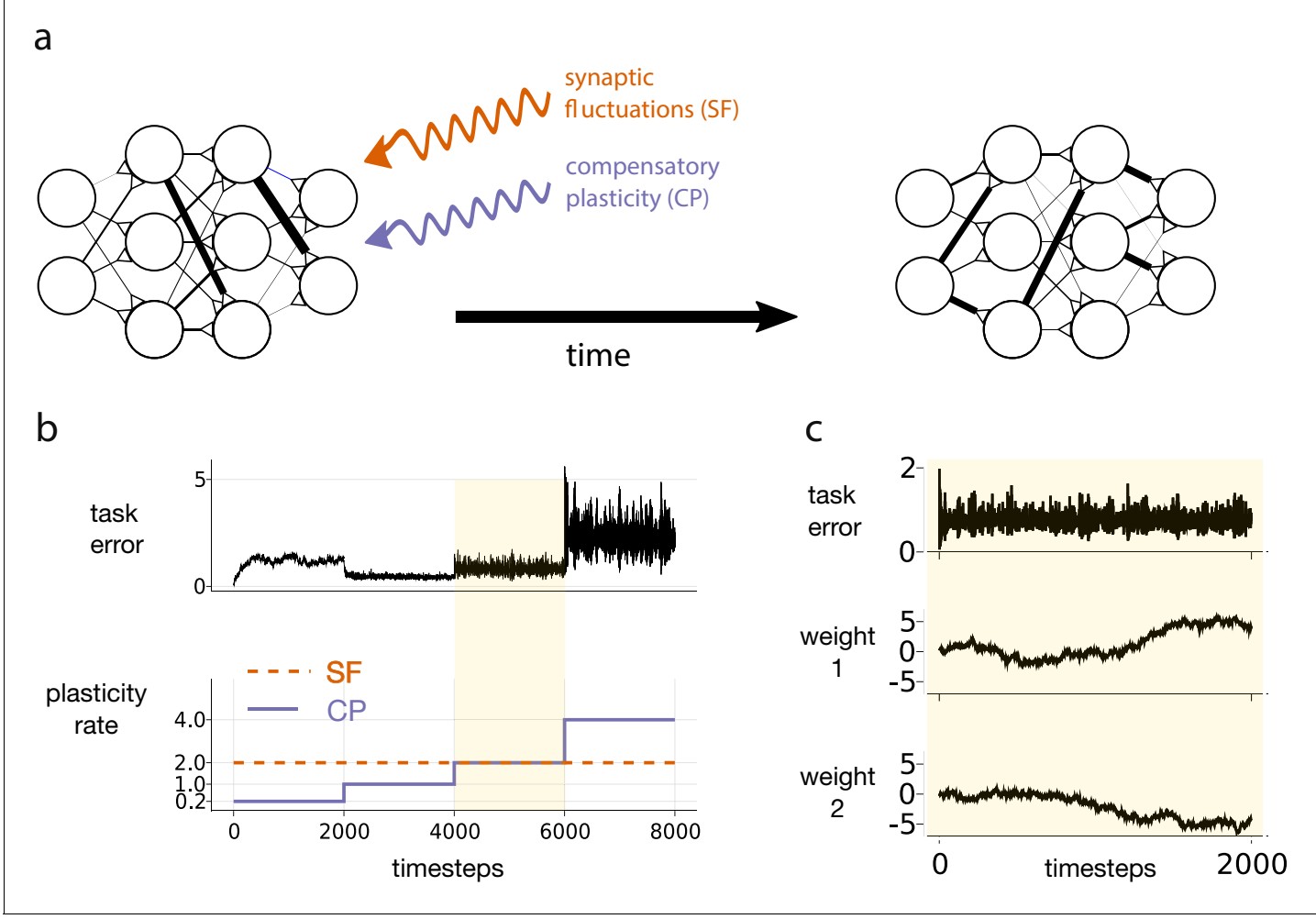

**Figure 1.** Motivating simulation results. (**a**) We consider a network attempting to retain previously learned information that is subject to ongoing synaptic changes due to synaptic fluctuations and compensatory plasticity. (**b**) Simulations performed in this study use an abstract, rate based neural network (described in section *Motivating example*). The rate of synaptic fluctuations is constant over time. By iteratively increasing the compensatory plasticity rate in steps we observe a 'sweet-spot' compensatory plasticity rate, which is lower than that of the synaptic fluctuations, and which best controls task error. (**c**) A snapshot of the simulation described in b, at the point where the rates of synaptic fluctuations and compensatory plasticity are matched. Even as task error fluctuates around a mean value, individual weights experience systematic changes.

- Compensatory plasticity $\Delta\mathbf{c}(t)$: We demand that compensatory plasticity mechanisms change the network state in a direction of decreasing task error, on average. As such, they cause the network to preserve previously stored information, though not in general by restoring synaptic weights to their previous values following a perturbation.

## Motivating example

Having described a generic modelling framework, we next uncover a key observation using a simple simulation.

*Figure 1* depicts an abstract, artificial neural network trying to maintain a given input-output mapping over time, which is analogous to preservation of a memory trace or learned task. At every timestep, synaptic fluctuations corrupt the weights, and a compensatory plasticity mechanism acts to reduce any error in the input-output mapping (see *Equation (1)*). We fix the rate (i.e. magnitude per timestep) of synaptic fluctuations throughout. We increase the compensatory plasticity rate in stages, ranging from a level far below the synaptic fluctuation rate, to a level far above it. Each stage is maintained so that task error can settle to a steady state.

## Box 1. Mathematical assumptions on plasticity.

To quantify memory quality/task performance we consider a loss function $F[\mathbf{w}(t^*)]$, which is twice differentiable in $\mathbf{w}(t)$. This loss function is simply an implicit measure of memory quality; we do not assume that the network explicitly represents $F$, or has direct access to it. Consider an infinitesimal weight-change $\Delta\mathbf{w}$ over the infinitesimal time-interval $\Delta t$. We apply a second order Taylor expansion to express the consequent change in task error: $\Delta F = F[\mathbf{w}(t^*) + \Delta\mathbf{w}] - F[\mathbf{w}(t^*)]$:

$$
\begin{aligned}
\Delta F &= \Delta\epsilon^T \nabla F[\mathbf{w}(t^*)] + \Delta\mathbf{c}^T \nabla F[\mathbf{w}(t^*)] \\
&+ \frac{1}{2}\Delta\mathbf{c}^T(\nabla^2 F[\mathbf{w}(t^*)])\Delta\mathbf{c} + \frac{1}{2}\Delta\epsilon^T(\nabla^2 F[\mathbf{w}(t^*)])\Delta\epsilon \\
&+ \Delta\mathbf{c}^T(\nabla^2 F[\mathbf{w}(t^*)])\Delta\epsilon + \mathcal{O}(\|\Delta\mathbf{c} + \Delta\epsilon\|_2^3).
\end{aligned}
\tag{2}
$$

Here, $\nabla F[\mathbf{w}(t^*)]$ and $\nabla^2 F[\mathbf{w}(t)]$ represent the first two derivatives (gradient and hessian) of $F[\mathbf{w}(t^*)]$, with respect to a change in the weights $\mathbf{w}(t^*)$. We assume that $\Delta\mathbf{c}$ and $\Delta\epsilon$ are sufficiently small (due to the short time interval) that the third-order term $\mathcal{O}(\|\Delta\mathbf{c} + \Delta\epsilon\|_2^3)$ can be ignored.

Next, we assume that $\Delta\mathbf{c}$ and $\Delta\epsilon$ are generated from unknown probability distributions. We place some constraints on these distributions. Firstly, synaptic fluctuations should be uncorrelated, in expectation, with the derivatives of $F[\mathbf{w}]$, which govern learning. Accordingly,

$$
\mathbb{E}[\Delta\epsilon^T \nabla F[\mathbf{w}(t^*)]\%] = 0,
\tag{3a}
$$

$$
\mathbb{E}[\Delta\mathbf{c}^T(\nabla^2 F[\mathbf{w}(t^{*\%})])\Delta\epsilon] = 0.
\tag{3b}
$$

Secondly, we require that $\Delta\mathbf{c}$ points in a direction of plasticity that decreases task error, for sufficiently small

$$
\|\Delta\mathbf{c}\|_2 \Delta\mathbf{c}^T \nabla F[\mathbf{w}(t^*)] < 0.
\tag{3c}
$$

Two interesting phenomena emerge. The task error of the network is smallest when the compensatory plasticity rate is smaller than the synaptic fluctuation rate (*Figure 1b*). Meanwhile, individual weights in the network continually change even as overall task error remains stable due to redundancy in the weight configuration (*Figure 1c*), (see e.g. *Rule et al., 2019* for a review).

In this simple simulation, we made a number of arbitrary and non-biologically motivated choices. In particular, we used an abstract, rate-based network, and synthesised compensatory plasticity directions using the biologically questionable backpropagation rule (see Materials and methods for full simulation details). Nevertheless, *Figure 1* highlights a phenomenon that we claim is more general:

*The 'sweet-spot' compensatory plasticity rate that leads to optimal, steady-state retention of previously learned information is at most equal to the rate of synaptic fluctuations, and often less.*

In the remainder of the results section, we will build intuition as to when and why this claim holds. We will also explore factors influence the precise 'sweet-spot' compensatory plasticity rate.

### The loss landscape

In order to analyse a general learning scenario that can accommodate biologically relevant assumptions about synaptic plasticity, we will develop a few general mathematical constructs that will allow us to draw conclusions about how synaptic weights affect the overall function of a network.

We first describe the 'loss landscape': a conceptually useful, geometrical visualisation of task error $F[\mathbf{w}]$ (see also *Figure 2*). Every point on the landscape corresponds to a different network state $\mathbf{w}$.

Whereas any point on a standard three-dimensional landscape has two lateral (xy) co-ordinates, any point on the loss landscape has $N$ co-ordinates representing each synaptic strength. Plasticity changes $\mathbf{w}$, and thus corresponds to movement on the landscape. Any movement $\Delta\mathbf{w}$ has both a direction $\hat{\Delta}\mathbf{w}$ (where hats denote normalised vectors), and a magnitude $\|\Delta\mathbf{w}\|_2$. Meanwhile, the elevation of a point $\mathbf{w}$ on the landscape represents the degree of task error, $F[\mathbf{w}]$. Compensatory plasticity improves task error, and thus moves downhill, regardless of the underlying plasticity mechanism.

## Understanding curvature in the loss landscape

Intuitively, one would expect task-independent synaptic fluctuations to increase task error. This is true even if fluctuations are unbiased in moving in an uphill or downhill direction on the loss landscape (see *Equation (3a)*) due to the curvature of the landscape (see *Figure 2C*). For instance, the slope (mathematically represented by the gradient $\nabla F[\mathbf{w}]$) at the bottom of a valley is zero. However, every direction is positively curved, and thus moves uphill. More generally, consider a fluctuation that is unbiased in selecting uphill or downhill directions, at a network state $\mathbf{w}$. The fluctuation will increase task error in expectation if the total curvature of the upwardly curved directions at $\mathbf{w}$ exceeds that of the downwardly curved directions, as illustrated in *Figure 2c*. We refer to such a state as partially trained. If all directions are upwardly curved, such as at/near the bottom of a valley, we refer to the state as highly trained. Mathematical definitions for these terms are provided in *Box 2*.

Comparison of the upward curvature of different plasticity directions plays an important role in the remainder of the section. Therefore, we introduce the following operator:

$$Q_{\mathbf{w}}[\mathbf{v}] = \hat{\mathbf{v}}^T \nabla^2 F[\mathbf{w}] \hat{\mathbf{v}}.$$

$Q_{\mathbf{w}}[\mathbf{v}]$ is mathematical shorthand for the degree of curvature in the direction $\mathbf{v}$, at point $\mathbf{w}$ on the loss landscape, and is depicted in *Figure 3a*. Note that $Q_{\mathbf{w}}[\mathbf{v}]$ depends solely upon the direction, and not the magnitude, of $\mathbf{v}$.

## An expression for the optimal degree of compensatory plasticity during learning

The rates of compensatory plasticity and synaptic fluctuations, at time $t$, are $\dot{\mathbf{c}}(t)$ and $\dot{\epsilon}(t)$, respectively. These rates may change continually over time. Let's temporarily assume they are fixed over a small time interval $[t, t + \Delta t]$. Thus,

$$\Delta\mathbf{c} = \dot{\mathbf{c}}(t)\Delta t \qquad\qquad \Delta\epsilon = \dot{\epsilon}(t)\Delta t. \qquad\qquad (5)$$

What magnitude of compensatory plasticity, $\|\Delta\mathbf{c}\|_2$, most decreases task error over $\Delta t$? The answer is

$$\|\Delta\mathbf{c}\|_2^* = \frac{-\Delta\hat{\mathbf{c}}^T \nabla\hat{F}[\mathbf{w}]}{Q_{\mathbf{w}}[\Delta\mathbf{c}]} \|\nabla F[\mathbf{w}]\|_2. \qquad\qquad (6)$$

A mathematical derivation is contained in *Box 3*, with geometric intuition in *Figure 3b*. Note that our answer turns out to be independent of the synaptic fluctuation rate $\dot{\epsilon}(t)$. Here,

- $\|\nabla F[\mathbf{w}]\|_2$ represents the sensitivity of the task error to changes (i.e. the steepness of the loss landscape).
- $-\Delta\hat{\mathbf{c}}^T \nabla\hat{F}[\mathbf{w}]$ represents the accuracy of the compensatory plasticity direction in conforming to the steepest downhill direction on the loss landscape (in particular, their normalised correlation).
- $Q_{\mathbf{w}}[\Delta\mathbf{c}]$ represents the upward curvature of the compensatory plasticity direction. As shown in *Figure 3b*, excessive plasticity in an upwardly curved, but downhill, direction, can eventually increase task error. Thus, upward curvature limits the ideal magnitude of compensatory plasticity in the direction $\Delta\hat{\mathbf{c}}$.

For now, *Equation (6)* is valid only if the compensatory plasticity direction is fixed during $\Delta t$. If we want *Equation (6)* to also be compatible with continually changing compensatory plasticity directions, it needs to be valid for an arbitrarily small $\Delta t$. However, enacting a non-negligible magnitude

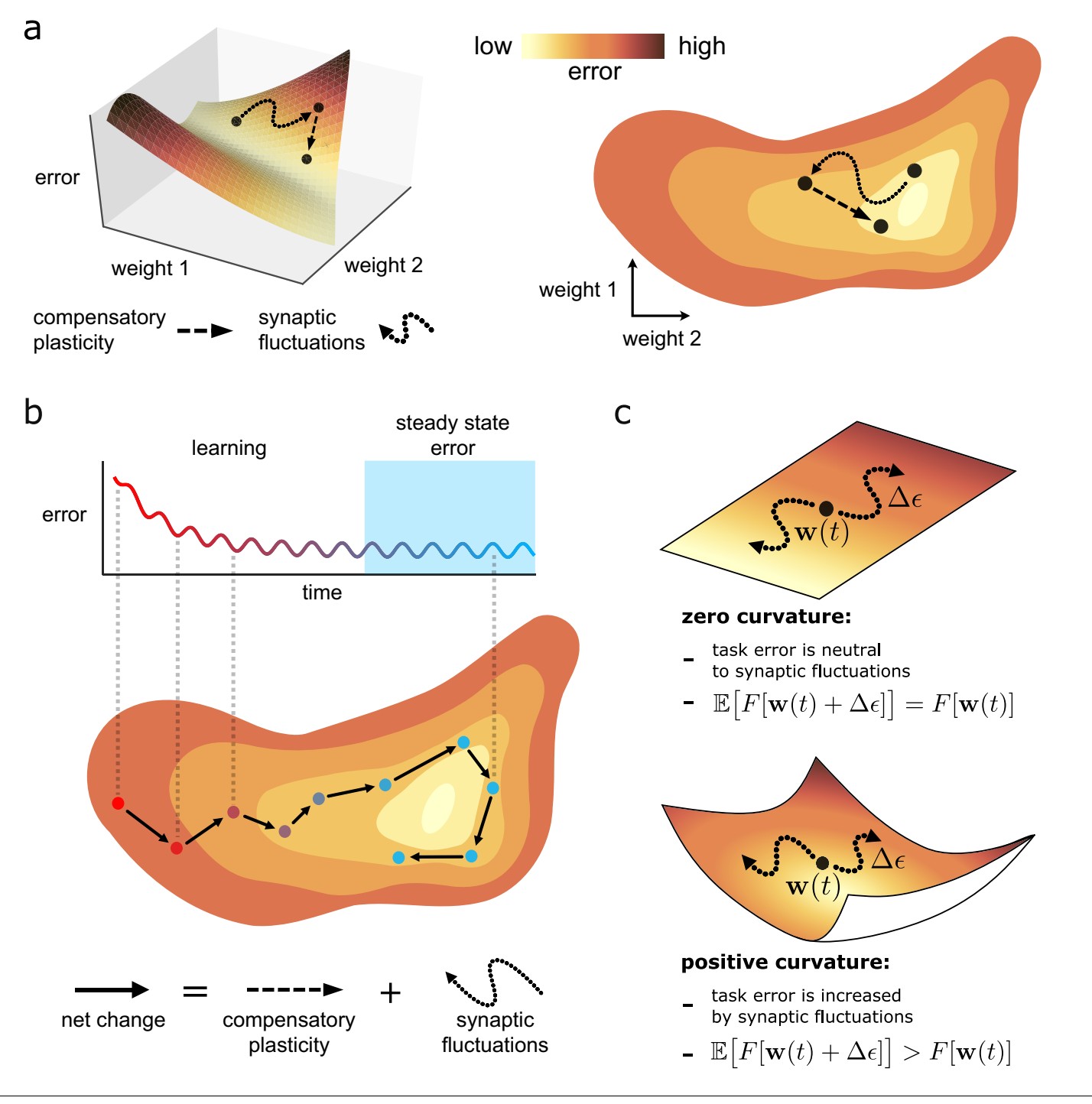

**Figure 2.** Task error landscape and synaptic weight trajectories. (a) Task error is visualised as the height of a 'landscape'. Lateral co-ordinates represent the values of different synaptic strengths (only two are visualisable in 3D). Any point on the landscape defines a network state, and the height of the point is the associated task error. Both compensatory plasticity and synaptic fluctuations alter network state, and thus task error, by changing synaptic strengths. Compensatory plasticity reduces task error by moving 'downwards' on the landscape. (b) Eventually, an approximate steady state is reached where the effect of the two competing plasticity sources on task error cancel out. The synaptic weights wander over a rough level set of the landscape. (c) The effect of synaptic fluctuations on task error depends on local curvature in the landscape. Top: a flat landscape without curvature. Even though the landscape is sloped, synaptic fluctuations have no effect on task error in expectation: up/downhill directions are equally likely. Bottom: Although up/downhill synaptic fluctuations are still equally likely, most directions are upwardly curved. Thus, uphill directions increase task error more, and downhill directions decrease task error less. So in expectation, synaptic fluctuations wander uphill.

## Box 2. Curvature and the loss landscape.

Consider a fluctuation $\Delta\mathbf{w}$ at a state $\mathbf{w}$. The change in task error, to second order, can be written as

$$F[\mathbf{w}+\Delta\mathbf{w}]-F[\mathbf{w}]\approx\Delta\mathbf{w}^T\nabla F[\mathbf{w}]+\frac{1}{2}\Delta\mathbf{w}^T\nabla^2 F[\mathbf{w}]\Delta\mathbf{w} \tag{4}$$

via a Taylor expansion. Suppose the fluctuation is task-independent. So it is unbiased with respect to selecting uphill/downhill, and more/less curved directions on the loss landscape. In this case

$$\mathbb{E}[\nabla F[\mathbf{w}]^T\Delta\mathbf{w}]=0$$

In expectation, *Equation (4)* thus becomes

$$\mathbb{E}[F[\mathbf{w}+\Delta\mathbf{w}]-F[\mathbf{w}]]=\|\Delta\mathbf{w}\|_2^2\frac{Tr(\nabla^2 F[\mathbf{w}])}{2N}.$$

If $Tr(\nabla^2 F[\mathbf{w}])>0$, then the expected change in task error is positive, and we refer to the network state as 'partially trained'. If additionally, $\nabla^2 F[\mathbf{w}]\succeq 0$, that is, $\Delta\mathbf{w}^T\nabla^2 F[\mathbf{w}]\Delta\mathbf{w}\geq 0$ for any choice of $\Delta\mathbf{w}$, then we refer to the network as highly trained. The 'highly trained' condition always holds in a neighbourhood of a local minimum of task error.

$\|\Delta\mathbf{c}\|_2^*$ of plasticity over an arbitrarily small time interval $\Delta t$ would require an unattainable, 'infinitely-fast' plasticity rate.

In fact, we show in the next section that our expression for $\|\Delta\mathbf{c}\|_2^*$ does become compatible with continuously changing plasticity at the end of learning, when task-error is stable.

### Characterising the optimal rate of compensatory plasticity at steady state

Consider a scenario where task error is approximately stable. In this case, $\Delta F\approx 0$ over $\Delta t$. In this scenario, *Equation (6)* simplifies to

$$\frac{\|\Delta\mathbf{c}\|_2^{*2}}{\|\Delta\epsilon\|_2^2}=\frac{Q_\mathbf{w}[\Delta\epsilon]}{Q_\mathbf{w}[\Delta\mathbf{c}]}, \tag{9a}$$

as derived in *Box 4* and illustrated geometrically in *Figure 3c*. We see that the magnitude $\|\Delta\mathbf{c}\|_2^*$ is proportional to $\|\Delta\epsilon\|_2$, which is itself proportional to $\Delta t$ from *Equation (5)*, given some fixed rate of synaptic fluctuations. Thus, $\|\Delta\mathbf{c}\|_2^*$ is attainable even as $\Delta t$ shrinks to zero, and is thus compatible with continually changing compensatory plasticity directions. In this case, *Equation (9)* can be rewritten as

$$\frac{\|\dot{\mathbf{c}}(t)\|_2^{*2}}{\|\dot{\epsilon}(t)\|_2^2}=\frac{Q_\mathbf{w}[\dot{\epsilon}(t)]}{Q_\mathbf{w}[\dot{\mathbf{c}}(t)]}. \tag{9b}$$

*Equation (9)* is a key result of the paper. It applies regardless of the underlying plasticity mechanisms that induced $\Delta\mathbf{c}$ and $\Delta\epsilon$. It is compatible with continually or occasionally changing directions of compensatory plasticity (i.e. infinitesimal or non-infinitesimal $\Delta t$). It says that the optimal compensatory plasticity rate, relative to the rate of synaptic fluctuations, depends on the relative upward curvature of these two plasticity directions on the loss landscape.

A corollary is that the optimal rate of compensatory plasticity is greater during learning than at steady state. If we substitute the steady-state requirement: $\mathbb{E}[\Delta F]=0$, with the condition for learning: $\mathbb{E}[\Delta F]<0$, in the derivation of *Box 4*, then we get

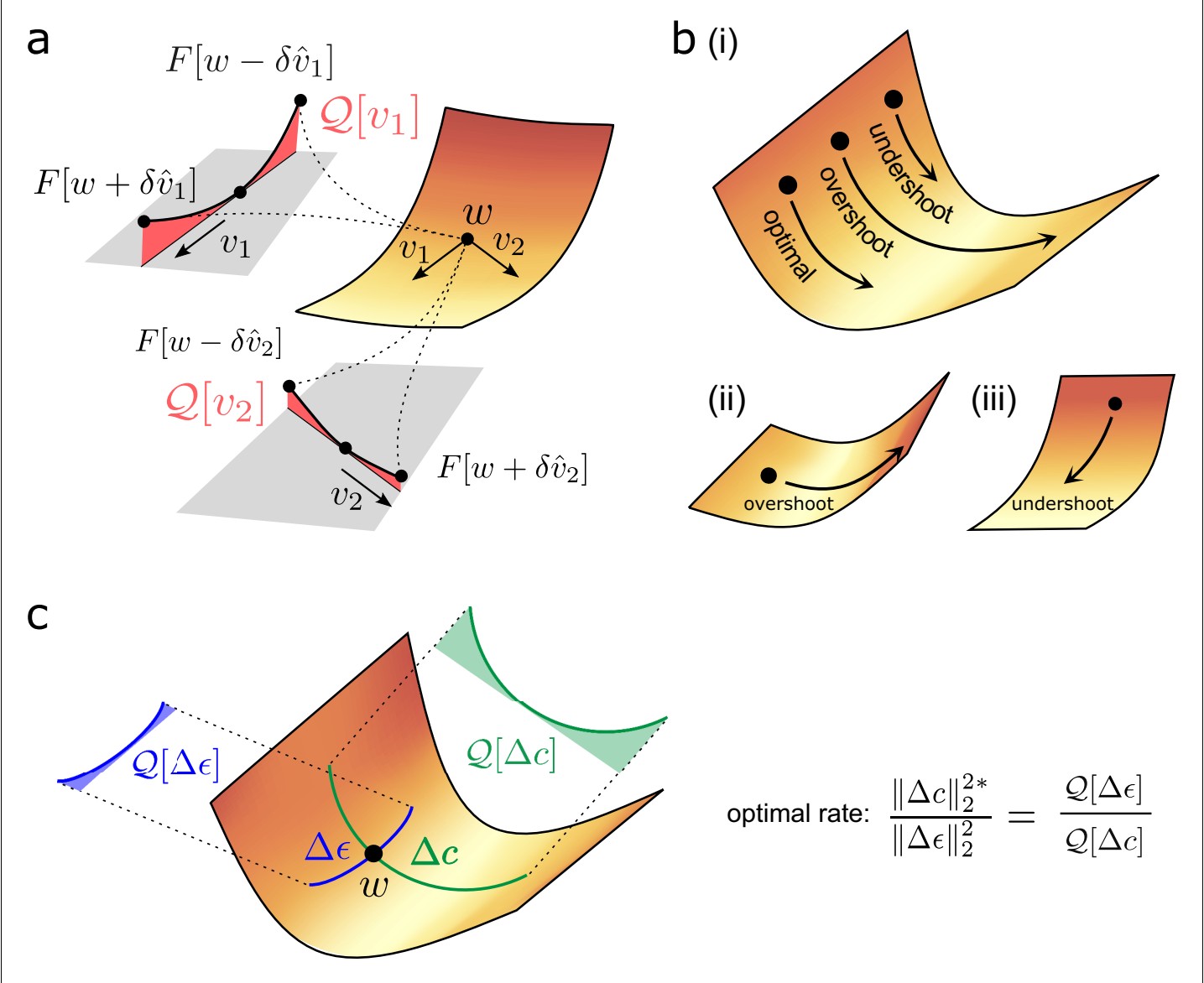

**Figure 3.** Quantifying effect of task error lanscape curvature on compensatory plasticity. (a) Geometrical intuition behind the operator $Q_{\mathbf{w}}$. The operator charts the degree to which a (normalised) direction is upwardly curved (i.e. lifts off the tangent plane depicted in grey). The red, shaded areas filling the region between the tangent plane and the upwardly curved directions are proportional to $Q_{\mathbf{w}}[v_1]$, and $Q_{\mathbf{w}}[v_2]$, respectively. (b) Compensatory plasticity points in a direction of locally decreasing task-error. Excessive plasticity in this direction can be detrimental, due to upward curvature ('overshoot'). The optimal magnitude for a given direction is smaller if upward curvature (i.e. the $Q$-value) is large, as for cases (i) and (ii), and if the initial slope is shallow, as for case (ii). It is greater if the initial slope is steep, as for case (iii). This intuition underlies *Equation (6)* for the optimal magnitude of a given compensatory plasticity direction, which includes as a coefficient the ratio of slope to curvature. (c) *Equation (11)* depends upon the ratio of the upward curvatures in the two plasticity directions, $\Delta c$, and $\Delta \epsilon$. As illustrated, steep downhill directions often exhibit more upward curvature than arbitrary directions. In such cases, the optimal magnitude of compensatory plasticity should be outcompeted by synaptic fluctuations.

$$\frac{\|\Delta \mathbf{c}\|_2^{*2}}{\|\Delta \epsilon\|_2^2} \geq \frac{Q_{\mathbf{w}}[\Delta \epsilon]}{Q_{\mathbf{w}}[\Delta \mathbf{c}]}. \tag{10}$$

Indeed, the faster the optimal potential learning rate $\mathbb{E}[\Delta F]$, the greater the optimal compensatory plasticity rate. Thus $\|\Delta \mathbf{c}\|_2^*$ decreases as learning slows to a halt, eventually reaching the level of *Equation (9b)*.

## Box 3. Optimal magnitude of compensatory plasticity.

Let us rewrite **Equation (2)**, using the operator $Q$ and omitting higher order terms, as justified in **Box 1**:

$$\Delta F = \Delta \epsilon^T \nabla F[\mathbf{w}(t^*)] + \Delta \mathbf{c}^T \nabla F[\mathbf{w}(t^*)]$$
$$+ \frac{1}{2}\|\Delta \mathbf{c}\|_2^2 Q_{\mathbf{w}(t^*)}[\Delta \mathbf{c}] + \frac{1}{2}\|\Delta \epsilon\|_2^2 Q_{\mathbf{w}(t^*)}[\Delta \epsilon] \tag{7}$$
$$+ \Delta \mathbf{c}^T (\nabla^2 F[\mathbf{w}(t^*)]) \Delta \epsilon.$$

We can substitute our assumptions on synaptic fluctuations (**Equations (3)**) into **Equation (7)** to get

$$\mathbb{E}[\Delta F] \tag{8}$$

Note that the requirement for assumption (3b) can be removed, but the alternative resulting derivation is more involved (see SI section two for this alternative).
We can differentiate **Equation (8)** in $\|\Delta \mathbf{c}\|_2$, to get:

$$\frac{d\mathbb{E}[\Delta F]}{d\|\Delta \mathbf{c}\|_2} = \Delta \hat{\mathbf{c}}^T \nabla F[\mathbf{w}(t^*)] + \|\Delta \mathbf{c}\|_2 Q_{\mathbf{w}(t^*)}[\Delta \mathbf{c}].$$

The root of this derivative gives a global minimum of the **Equation (8)** in $\|\Delta \mathbf{c}\|_2$, as long as $Q_{\mathbf{w}(t^*)}[\Delta \mathbf{c}] \geq 0$ holds (justified in SI section 2.1). We get **Equation (6)**, which defines the compensatory plasticity magnitude that minimises $\Delta F$, and thus overall task error, at time $t^* + \Delta t$.

## Main claim

We now claim that generically, the optimal compensatory plasticity rate should not outcompete the rate of synaptic fluctuations at steady state error. We will first provide geometric intuition for our claim, before bolstering with analytical arguments and making precise our notion of 'generically'.

## Box 4. Optimal compensatory plasticity magnitude at steady state error.

Let us substitute the special condition $\mathbb{E}[\Delta F] = 0$ (steady-state task error) into **Equation (8)**. This gives

$$0 = \Delta \mathbf{c}^T \nabla F[\mathbf{w}(t^*)] + \frac{1}{2}\|\Delta \mathbf{c}\|_2^2 Q_{\mathbf{w}(t^*)}[\Delta \mathbf{c}] + \frac{1}{2}\|\Delta \epsilon\|_2^2 Q_{\mathbf{w}(t^*)}[\Delta \epsilon].$$

Next, we substitute in our optimal reconsolidation magnitude (**Equation (6)**). This gives

$$0 = -\frac{1}{2}\|\Delta \mathbf{c}\|_2^2 Q[\Delta \mathbf{c}] + \frac{1}{2}\|\Delta \epsilon\|_2^2 Q[\Delta \epsilon],$$

which in turn implies the result (**Equation (9)**).
Note that **Equation (9)** is only valid when both the numerator and denominator of the right hand side are both positive. The converse is unlikely in a partially trained network, and impossible in a highly trained network (see SI section 2.1).

From *Equation (9)*, our main claim holds if

$$Q_{\mathbf{w}}[\Delta \mathbf{c}] \geq Q_{\mathbf{w}}[\Delta \epsilon], \tag{11}$$

that is, $\Delta \mathbf{c}$ points in a more upwardly curved direction than $\Delta \epsilon$. When would this be true?

First consider $\Delta \epsilon$. Statistical independence from the task error means it should point in an 'averagely' curved direction. Mathematically (see SI secton 2.1), this means

$$\mathbb{E}[Q_{\mathbf{w}}[\Delta \epsilon]] = \frac{Tr(\nabla^2 F[\mathbf{w}])}{N}. \tag{12}$$

Our assumption of 'average' curvature fails if synaptic fluctuations are specialised to 'unimportant' synapses whose changes have little effect on task error. In this case $Q_{\mathbf{w}}[\Delta \epsilon]$ would be even smaller, since $\Delta \epsilon$ would be constrained to consistently shallow, less-curved directions. Thus, this possibility does not interfere with our main claim.

For *Equation (11)* to hold, $\Delta \mathbf{c}$ should point in directions of 'more-than-average' upward curvature. This follows intuitively because a steep downhill direction, which effectively reduces task error, will usually have higher upward curvature than an arbitrary direction (see *Figure 3c* for intuition). It remains to formalise this argument mathematically, and consider edge cases where it doesn't hold.

## Dependence of the optimal magnitude of steady-state, compensatory plasticity on the mechanism

Compensatory plasticity is analogous to learning, since it acts to reduce task error. We do not yet know the algorithms that neural circuits use to learn, although constructing biologically plausible learning algorithms is an active research area. Nevertheless, all the potential learning algorithms we are aware of fit into three broad categories. For each category, we shall show why and when our main claim holds. We will furthermore investigate quantitative differences in the optimal compensatory plasticity rate, across and within categories. A full mathematical justification of all the assertions we make is found in SI section 1.3.

We first highlight a few general points:

- For any compensatory plasticity mechanism, $Q_{\mathbf{w}}[\Delta \mathbf{c}]$ depends not only on the algorithm, but the point $\mathbf{w}$ on the landscape. We cannot ever claim that *Equation (11)* holds for all network states.
- We calculate the expected value of $Q_{\mathbf{w}}[\Delta \mathbf{c}]$ for an 'average', trained, state $\mathbf{w}$, across classes of algorithm. This corresponds to a plausible best-case tuning of compensatory plasticity that a neural circuit might be able to achieve. Any improvement would rely on online calculation of $Q_{\mathbf{w}}[\Delta \mathbf{c}]$, which we do not believe would be plausible biologically.

Learning algorithms attempt to move to the bottom of the loss landscape. But they are blind. Spying a distant valley equates to 'magically' predicting that a very different network state will have very low task error. How do they find their way downhill? There are three broad strategies (*Raman and O'Leary, 2021*):

- $0^{\text{th}}$ order algorithms take small, exploratory steps in random directions. Information from the change in task error over these steps informs retained changes. For instance, steps that improve task error are retained. A notable 0-order algorithm is REINFORCE (*Williams, 1992*). Many computational models of biological learning in different circuits derive from this algorithm (*Seung, 2003*; *Fee and Goldberg, 2011*; *Bouvier et al., 2018*; *Kornfeld et al., 2020*).
- $1^{\text{st}}$ order algorithms explicitly approximate/calculate, and then step down the locally steepest direction (i.e. the gradient $\nabla F[\mathbf{w}]$). The backpropagation algorithm implements perfect

**Table 2.** Table elements highlighted in teal correspond to scenarios in which our main claim holds, as *Equation (11)* is satisfied.

| | Quadratic $F[\mathbf{w}]$ | Nonlinear $F[\mathbf{w}]$, low steady-state error | Nonlinear $F[\mathbf{w}]$, high steady-state error |
|---|---|---|---|
| $0^{\text{th}}$ order algorithm | $Q[\Delta \mathbf{c}] \approx Q[\Delta \epsilon]$ | $Q[\Delta \mathbf{c}] \approx Q[\Delta \epsilon]$ | $Q[\Delta \mathbf{c}] \approx Q[\Delta \epsilon]$ |
| $0^{\text{st}}$ order algorithm | $Q[\Delta \mathbf{c}] \geq Q[\Delta \epsilon]$ | $Q[\Delta \mathbf{c}] \geq Q[\Delta \epsilon]$ | $Q[\Delta \mathbf{c}] \approx Q[\Delta \epsilon]$ |
| $0^{\text{nd}}$ order algorithm | $Q[\Delta \mathbf{c}] \approx Q[\Delta \epsilon]$ | $Q[\Delta \mathbf{c}] \approx Q[\Delta \epsilon]$ | $Q[\Delta \mathbf{c}] \leq Q[\Delta \epsilon]$ |

gradient descent. Many approximate gradient descent methods with more biologically plausi-ble assumptions have been developed in the recent literature (see e.g. *Murray, 2019*; *Whittington and Bogacz, 2019*; *Bellec et al., 2020*; *Lillicrap et al., 2016*; *Guerguiev et al., 2017*, and *Lillicrap et al., 2020* for a review).

- 2nd order algorithms additionally approximate/calculate the hessian $\nabla^2 F[\mathbf{w}]$, which provides information on local curvature. They look for descent directions that are both steep, and less upwardly curved. We doubt it is possible for biologically plausible learning rules to accurately

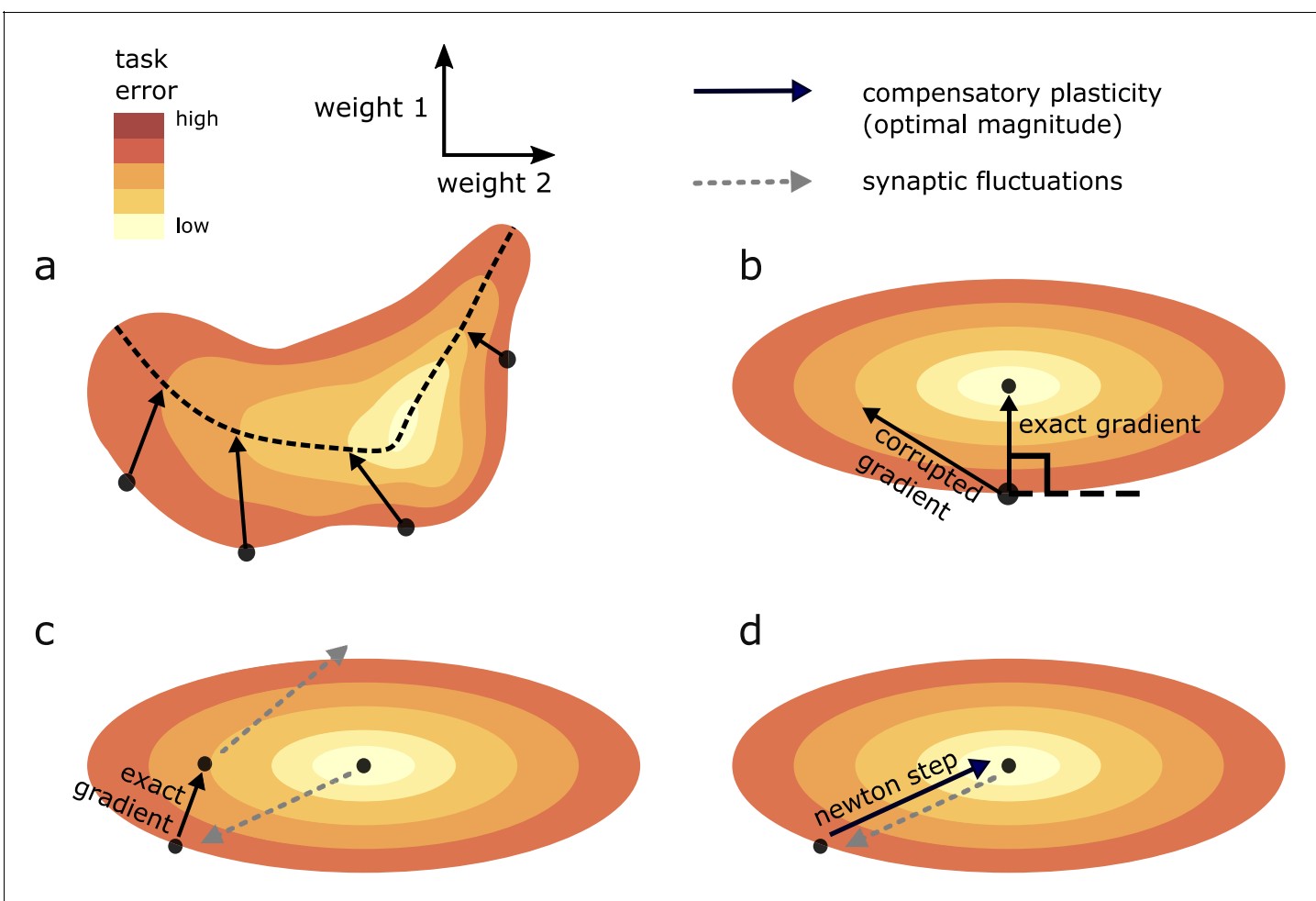

**Figure 4.** Geometric intuition for the optimal magnitude of different compensatory plasticity directions. Colours depict level sets of the loss landscape. Elliptical level sets correspond to a quadratic loss function (which approximates any loss function in the neighbourhood of a local minimum). In c and d, we depict compensatory plasticity and synaptic fluctuations as sequential, alternating processes for illustrative purposes, although they are modelled as concurrent throughout the paper. (**a**) Compensatory plasticity directions locally decrease task error, so point from darker to lighter colours. Optimal magnitude is reached when the vectors 'kiss' a smaller level set, that is, intersect that level set while running parallel to its border. Increasing magnitude past this past this point increases task error, by moving network state to a higher-error level set. (**b**) If compensatory plasticity is parallel to the gradient (i.e. it enacts gradient descent), then it runs perpendicular to the border of the level set on which it lies (i.e. the tangent plane). This is shown explicitly for the 'exact gradient' direction of plasticity. The optimal magnitude of plasticity in this direction is smaller than that of a corrupted gradient descent direction, even though the former is more effective in reducing task error, because the exact gradient points in a more highly curved direction. (**c**) Synaptic fluctuations of a certain magnitude perturb the network state. The optimal magnitude of compensatory plasticity (in the exact gradient descent direction, for this example) is significantly smaller than that of the synaptic fluctuations, using the geometric heuristic explained in (**a**). If the magnitude of compensation increased to match the synaptic fluctuation magnitude there would be overshoot, and task error would converge to a higher steady state. (**d**) If compensatory plasticity mechanisms can perfectly calculate both the local gradient and hessian (curvature) of the loss landscape, then network state will move in the direction of the 'Newton step'. In the quadratic case (elliptical level sets), this will directly 'backtrack' the synaptic fluctuations. Thus, the optimal magnitude of compensatory plasticity will be equal to that of the synaptic fluctuations. However, time delays in the sensing of synaptic fluctuations and limited precision of the compensatory plasticity mechanism will preclude this.

approximate the hessian, which has $N^2$ entries representing the interaction between every possible pair of synaptic weights.

*Table 2* shows the categories for which our main claim holds.

We first consider the simplest case of a quadratic loss function $F[\mathbf{w}]$. Here, directions of curvature in any direction are constant (mathematically, the hessian $\nabla^2 F[\mathbf{w}]$ does not vary with network state). Moreover, the gradient obeys a consistent relationship with the hessian:

$$\nabla F[\mathbf{w}] = \nabla^2 F[\mathbf{w}^*](\mathbf{w} - \mathbf{w}^*). \tag{13}$$

Components of $(\mathbf{w} - \mathbf{w}^*)$ with high upward curvature are magnified under the transformation $\nabla^2 F[\mathbf{w}^*]$, since they correspond to eigenvectors of $\nabla^2 F[\mathbf{w}^*]$ with high eigenvalue. Conversely, components with low upward curvature are shrunk. As the gradient $\nabla F[\mathbf{w}]$ is the output of such a transformation from *Equation (13)*, this suggests it is biased towards directions of high upward curvature. Indeed, we can quantify this bias. Let $\{\lambda_i\}$ be the eigenvalues of $\nabla^2 F[\mathbf{w}^*]$, and $\{c_i\}$ the projections of the corresponding eigenvectors onto $\mathbf{w} - \mathbf{w}^*$. Then

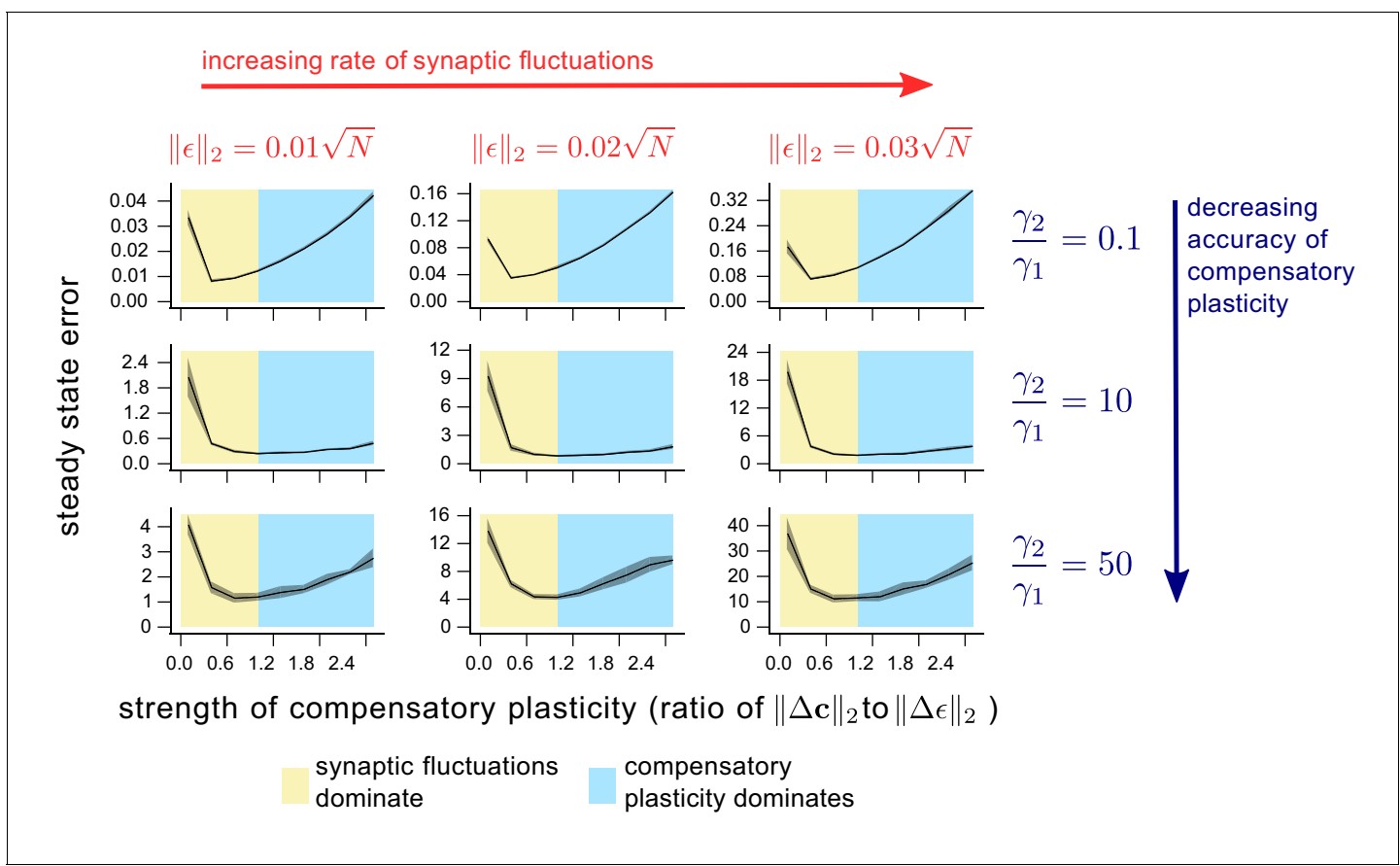

**Figure 5.** The dependence of steady state task performance in a nonlinear network on the magnitudes of compensatory plasticity and synaptic fluctuations, and on the learning rule quality. Each $(x, y)$ value on a given graph corresponds to an 8000 timepoint nonlinear neural network simulation (see 'Methods' for details). The $y$ value gives the steady-state task error (average task error of the last 500 timepoints) of the simulation, while the $x$ value gives the ratio of the magnitudes of the compensatory plasticity and synaptic fluctuations terms. Steady state error is averaged across 8 simulation repeats; shading depicts one standard deviation. Between graphs, we change simulation parameters. Down rows, we increase the proportionate noise corruption of the compensatory plasticity term (see Materials and methods section for details). Across columns, we increase the magnitude of synaptic fluctuations.

The online version of this article includes the following figure supplement(s) for figure 5:

**Figure supplement 1.** The dependence of steady state task performance in a linear network on the magnitudes of compensatory plasticity and synaptic fluctuations, and on the learning rule quality.

$$Q_{\mathbf{w}}[\nabla F[\mathbf{w}]] = \frac{\sum_{i=1}^{N} c_i^2 \lambda_i^3}{\sum_{i=1}^{N} c_i^2 \lambda_i^2}. \tag{14}$$

The value of **Equation (14)** depends on the values $\{c_i\}$. In the 'average' case, where they are equal, and $\mathbf{w} - \mathbf{w}^*$ is thus a direction of 'average' curvature, $Q_{\mathbf{w}}[\nabla F[\mathbf{w}]] \geq Q_{\mathbf{w}}[\Delta\epsilon]$ holds. This inequality gap widens with increasing anisotropy in the curvature of different directions (i.e. with a wider spread of eigenvalues $\lambda_i$, corresponding to more elliptical/less circular level sets in the illustration of **Figure 4b**). Indeed, simulation results in **Figure 5—figure supplement 1** (top row) show how the ratio $\|\Delta\mathbf{c}\|_2 : \|\Delta\epsilon\|_2$ that optimises steady-state task error is significantly less than one, in a quadratic error function where compensatory plasticity accurately follows the gradient, and for different synaptic fluctuation rates.

What about the case of a nonlinear loss function? Close to a minimum $\mathbf{w}^*$, the relationship of **Equation (13)** approximately holds (the loss function is locally quadratic). So if steady-state error is very low, we can directly transport the intuition of the quadratic case. However when steady state error increases, **Equation (13)** becomes increasingly approximate. In the limiting case, we could consider $\nabla F[\mathbf{w}]$ as being completely uncorrelated from $\nabla^2 F[\mathbf{w}]$, in which case $Q_{\mathbf{w}}[\nabla F[\mathbf{w}]] \approx Q_{\mathbf{w}}[\Delta\epsilon]$ would hold. Numerical results in **Figure 5** supports this assertion in nonlinear networks: the optimal ratio satisfies $\|\Delta\mathbf{c}\|_2 : \|\Delta\epsilon\|_2 \approx 1$ in conditions where steady-state task error is high, and $\|\Delta\mathbf{c}\|_2 : \|\Delta\epsilon\|_2 \leq 1$ in conditions where it is low.

Overall, we see that if $\Delta\mathbf{c} \propto -\nabla F[\mathbf{w}]$ (i.e. compensatory plasticity enacts gradient descent), then we would expect compensatory plasticity to be outcompeted by synaptic fluctuations to maintain optimal steady-state error.

Even if compensatory plasticity does not move in the steepest direction of error decrease (i.e. the error gradient), it must move in an approximate downhill direction to improve task error (see e.g. **Raman et al., 2019**). Furthermore, the worse the quality of the gradient approximation, the *larger* the optimal level of compensatory plasticity (illustrated conceptually in **Figure 4b–c**, and numerically in **Figure 5** and **Figure 5—figure supplement 1**). Why? We can rewrite such a learning rule as

$$\Delta\mathbf{c} \propto -\nabla F[\mathbf{w}] + \nu,$$

where $\nu$ represents systematic error in the gradient approximation. The upward curvature in the direction $\Delta\mathbf{c}$ becomes a (nonlinear) interpolation of the upward curvatures in the directions $\nabla F[\mathbf{w}]$ and $\nu$ (see **Equation (A6)** of the SI). As long as $\nu$ is less biased towards high curvature directions than $\nabla F[\mathbf{w}]$ itself, then this decreases the upward curvature in the direction $\Delta\hat{\mathbf{c}}$, and thus increases the optimal compensatory plasticity rate. Indeed **Figure 5** shows in simulation that this rate increases for more inaccurate compensatory plasticity mechanisms.

We now turn to zero-order learning algorithms, such as REINFORCE. These do not explicitly approximate a gradient, but generate random plasticity directions, which are retained/opposed based upon their observed effect on task error. We would expect randomly generated plasticity directions to have 'average' upward curvature, similarly to synaptic fluctuations. In this case, we would therefore get $Q_{\mathbf{w}}[\Delta\mathbf{c}] \approx Q_{\mathbf{w}}[\Delta\epsilon]$, and compensatory plasticity should thus equal synaptic fluctuations in magnitude.

Finally, we consider second-order learning algorithms, and in particular the Newton update:

$$\nabla^2 F[\mathbf{w}]\Delta\mathbf{c} = -\nabla F[\mathbf{w}].$$

As previously discussed, we assume that learning algorithms that require detailed information about the Hessian are biologically implausible. As such, our treatment is brief, and mainly contained in SI section 2.2.2.

In a linear network, the Newton update corresponds to compensatory plasticity making a direct 'beeline' for $\mathbf{w}^*$ (see **Figure 4d**). As such $Q_{\mathbf{w}}[\Delta\mathbf{c}] = Q_{\mathbf{w}}[\Delta\epsilon]$ and the optimal magnitude of compensatory plasticity should match synaptic fluctuations. The same is true for a nonlinear network in a near-optimal state. However if steady-state task error is high in a nonlinear network, then compensatory plasticity should outcompete synaptic fluctuations. This case does not contradict our central claim however, since high task error at steady state implies that the task is not truly learned.

Together our results and analyses show that the magnitude of compensatory plasticity, at steady state task error, should be less or equal to that of synaptic fluctuations. This conclusion does not depend upon circuit architecture, or choice of biologically plausible learning algorithm.

## Discussion

A long-standing question in neuroscience is how neural circuits maintain learned memories while being buffeted by synaptic fluctuations from noise and other task-independent processes (*Fusi et al., 2005*). There are several hypotheses that offer potential answers, none of which are mutually exclusive. One possibility is that fluctuations only occur in a subset of volatile connections that are relatively unimportant for learned behaviours (*Moczulska et al., 2013*; *Chambers and Rumpel, 2017*; *Kasai et al., 2003*). Following this line of thought, circuit models have been proposed that only require stability in a subset of synapses for stable function (*Clopath et al., 2017*; *Mongillo et al., 2018*; *Susman et al., 2018*).

Another hypothesis is that any memory degradation due to fluctuations is counteracted by restorative plasticity processes that allow circuits to continually 'relearn' stored associations. The information source directing this restorative plasticity could come from an external reinforcement signal (*Kappel et al., 2018*), from interactions with other circuits (*Acker et al., 2018*), or spontaneous, network-level reactivation events (*Fauth and van Rossum, 2019*). A final possibility is that ongoing synaptic fluctuations are accounted for by behavioural changes unrelated to learned task performance .

All these hypotheses share two core assumptions that we make, and several include a third that our results depend on:

1. Not all synaptic changes are related to learning.
2. Unchecked, these learning-independent plasticity sources generically hasten the degradation of previously stored information within a neural circuit.
3. Some internal compensatory plasticity mechanism counteracts the degradation of previously stored information.

We extracted mathematical consequences of these three assumptions by building a general framework. We first modelled the the degree of degradation of previously learned information in terms of an abstract, scalar-valued, task error function or 'loss landscape'. The brain may not have, and in any case does not require, explicit representation of such a function for a specific task. All that is required is error feedback from the environment and/or some internal prediction.

We then noted that compensatory plasticity should act to decrease task error, and thus point in a downhill direction on the 'loss landscape'. We stress that we do not assume a gradient-based learning rule such as the backpropagation algorithm, the plausibility of which is an ongoing debate (*Whittington and Bogacz, 2019*).

Our results do not depend on whether synaptic changes during learning are gradual, or occur in large, abrupt steps. Although most theory work assumes plasticity to be gradual, there is evidence that plasticity can proceed in discrete jumps. For instance, abrupt potentiation of synaptic inputs that lead to the formation of place fields in mouse CA1 hippocampal neurons can occur within seconds as an animal explores a new environment (*Bittner et al., 2017*). Even classical plasticity paradigms that depend upon millisecond level precision in the relative timing of pre/post synaptic spikes follow a paradigm where there is a short 'induction phase' of a minute or so, following which there is a large and sustained change in synaptic efficacy (e.g. *Markram et al., 1997*; *Bi and Poo, 1998*). It is therefore an open question as to whether various forms of synaptic plasticity are best accounted for as an accumulation of small changes or a threshold phenomenon that results in a stepwise change. Our analysis is valid in either case. We quantify plasticity rate by picking a (large or small) time interval over which the net plasticity direction is approximately constant, and evaluate the optimal, steady-state magnitude of compensatory plasticity over this interval, relative to the magnitude of synaptic fluctuations.

A combination of learning-induced and learning-independent plasticity should lead to an eventual steady state level of task error, at which point the quality of stored information does not decay appreciably over time. The absolute quality of this steady state depends upon both the magnitude of the synaptic fluctuations, and the effectiveness of the compensatory plasticity.

Our main finding was that the quality of this steady state is optimal when the rate of compensatory plasticity does not outcompete that of the synaptic fluctuations. This result, which is purely mathematical in nature, is far from obvious. While it is intuitively clear that retention of circuit function will suffer when compensatory plasticity is absent or too weak, it is far less intuitive that the same is true generally when compensatory plasticity is too strong.

We also found that the precision of compensatory plasticity influenced its optimal rate. When 'precision' corresponds to the closeness of an approximation to a gradient-based compensatory plasticity rule, an increase in precision resulted in the optimal rate of compensatory plasticity being strictly less than that of fluctuations. In other words, sophisticated learning rules need to do less work to optimally overcome the damage done by learning-independent synaptic fluctuations. Indeed experimental estimates (see *Table 1*) suggest that activity-independent synaptic fluctuations can significantly outcompete systematic, activity-dependent changes in certain experimental contexts. Tentatively, this means that the high degree of synaptic turnover in these systems is in fact evidence for the operation of precise synaptic plasticity mechanisms as opposed to crude and imprecise mechanisms.

Our results are generic, in that they follow from fundamental mathematical relationships in optimisation theory, and hence are not dependent on particular circuit architectures or plasticity mechanisms. We considered cases in which synaptic fluctuations were distributed across an entire neural circuit. However, the basic framework easily extends, allowing for predictions in more specialised cases. For instance, recent theoretical work (*Clopath et al., 2017*; *Mongillo et al., 2018*; *Susman et al., 2018*) have hypothesised that synaptic fluctuations could be restricted to 'unimportant' synapses. These correspond to low curvature (globally insensitive) directions in the 'loss landscape'. Our framework (*Equation (9)* in particular) immediately predicts that the optimal rate of compensatory plasticity will decrease proportionately with this curvature.

Precise experimental isolation/elimination of the plasticity sources attributable to learning and retention of memories remains challenging. Nevertheless, in conventional theories of learning (e.g. Hebbian learning), neural networks learn through plasticity induced by patterns of pre- and postsynaptic neural activity. A reasonable approximation, therefore, is to equate the 'compensatory/learning-induced' plasticity of our paper with 'activity-dependent' plasticity in experimental setups. With this assumption, our results provide several testable predictions.

Firstly, our results show that that the rate of compensatory (i.e. learning-dependent) plasticity is greater when a neural circuit is in a phase of active learning, as opposed to maintaining previously learned information (see *Equation (10)* and the surrounding discussion). Consequently, the relative contribution of synaptic fluctuations to the overall plasticity rate should be lower in this case. It would be interesting to test whether this were indeed the case, by comparing brain circuits in immature vs mature organisms, and in neural circuits thought to be actively learning vs those thought to be retaining previously learned information. One way to do this would be to measure the covariance of functional synaptic strengths at coinnervated synapses using EM reconstructions of neural tissue. A higher covariance implies a lower proportion of activity-dependent (i.e. compensatory) plasticity, since co-innervated synapses share presynaptic activity histories. Interestingly, two very similar experiments (*Bartol et al., 2015*) and (*Dvorkin and Ziv, 2016*) did indeed examine covariance in EM reconstructions of hippocampus and neocortex, respectively. This covariance appears to be much lower in hippocampus (compare Figure 1 of *Bartol et al., 2015* to Figure 8 of *Dvorkin and Ziv, 2016*). Many cognitive theories characterise hippocampus as a continual learner and neocortex as a consolidator of previously learned information (e.g. *O'Reilly and Rudy, 2001*). Our analysis provides support for this hypothesis at a mechanistic level by linking low covariance in coinnervated hippocampal synapses to continual learning.

Secondly, a number of experimental studies (*Nagaoka et al., 2016*; *Quinn et al., 2019*; *Yasumatsu et al., 2008*; *Minerbi et al., 2009*; *Dvorkin and Ziv, 2016*) note a persistence of the bulk of synaptic plasticity in the absence of activity-dependent plasticity or other correlates of an explicit learning signal, as explained in our review of key experimental findings. However, there are two important caveats for relating our work to these experimental observations:

- Experimentally isolating different plasticity mechanisms, measuring synaptic changes, and accounting for confounding behavioural/physiological changes is extremely challenging. The most compelling in vivo support comes from *Nagaoka et al., 2016*, where an analogue of

compensatory plasticity in the mouse visual cortex was suppressed both chemically (by suppression of spiking activity) and behaviourally (by raising the mouse in visually impoverished conditions). Synaptic turnover was reduced by about half for both suppression protocols, and also when they were applied simultaneously. Further studies that quantified changes in synaptic strength in addition to spine turnover in an analogous setup would lend further credence to our results.

- We do not know if observed synaptic plasticity in the experiments we cite truly reflect a neural circuit attempting to minimise steady-state error on a particular learning goal (as captured through an abstract, implicit, 'loss function'). Our analysis simply shows that somewhat surprising levels of ongoing plasticity can be explained parsimoniously in such a framework. In particular, the concepts of 'learning' and behaviour have no clear relationship with neural circuit dynamics in vitro. Nevertheless, we might speculate that synapses could tune the extent to which they respond to 'endogenous' (task independent) signals versus external signals that could convey task information in the intact animal. Even if the information conveyed by activity-dependent signals were disrupted in vitro, the fact that activity-dependent signals determined such a small proportion of plasticity is notable, and seems to carry over to the in vivo case.

Thus, while our results offer a surprising agreement with a number of experimental observations, we believe it is important to further replicate measurements of synaptic modification in a variety of settings, both in vivo and in vitro. We hope our analysis provides an impetus for this difficult experimental work by offering a first-principles theory for the volatility of connections in neural circuits.

## Materials and methods

### Simulations

We simulated two types of network, which we refer to as linear (*Figure 5—figure supplement 1*) and nonlinear (*Figures 1* and *5*) respectively. We ran our simulations in the Julia programming language (version 1.3), and in particular used the Flux.jl software package (version 0.9) to construct and update networks. Source code is available at https://github.com/Dhruva2/OptimalPlasticityRatios (copy archived at swh:1:rev:fcb1717a822f90b733c49d62bfc2f970155b7364, *Raman, 2021*).

### Nonlinear networks

Networks were rate-based, with the firing rate $r(t)$ of a given neuron defined as

$$r(t) = \sigma(w^T(t)u(t)),$$

where $w$ is the vector of presynaptic strengths, $u$ represents the firing rate of the associated presynaptic neurons, and $\sigma(x) := \frac{1}{1+\exp(-x)}$ is the sigmoid function. Initial weight values were generated randomly, according to the standard Xavier distribution (*Glorot and Bengio, 2010*). Networks were organised into three layers, containing 12, 20, and 10 neurons, respectively. Any given neuron was connected to all neurons in the previous layer. For the first layer, the firing rates of the 'previous layer' corresponded to the network inputs.

### Linear networks

Networks were organised into an input layer of 12 neurons, and an output layer of 10 neurons. Each output neuron was connected to all input layer neurons. Networks were rate-based, with the firing rate $r(t)$ of a given neuron defined as

$$r(t) = w^T(t)u(t),$$

where $u_i(t)$ corresponds to the $i^{th}$ input (input-layer neuron) or the firing rate of the $i^{th}$ input-layer neuron (output-layer neuron). Initial weight values were generated randomly, according to the Xavier distribution (*Glorot and Bengio, 2010*).

### Task error

For each network, we generated 1000 different, random, input vectors. Each component of the vector was generated from a unit Gaussian distribution. Task error, at the $t^{th}$ timestep, was taken as the

mean squared error of the network in recreating the outputs of the initial ($t = 0$) network, in response to the suite of inputs. Mathematically, this equates to

$$F[\mathbf{w}(t)] = \frac{1}{|\mathcal{U}|}\sum_{u \in \mathcal{U}} \|y(\mathbf{w}(t), u) - y(\mathbf{w}(0), u)\|_2^2,$$

where $y(\mathbf{w}(t), u)$ denotes the output of the network given the synaptic strengths at time $t$, in response to input $u \in \mathcal{U}$. Note that this task error recreates the 'student-teacher' framework of e.g. (*Levin et al., 1990*; *Seung et al., 1992*), where a fixed copy of the initial network is the teacher.

## Weight dynamics

At each simulation timestep, synaptic weights were updated as

$$\Delta\mathbf{w}_{t+1} = \Delta\mathbf{c}_t + \Delta\epsilon_t.$$

We took the synaptic fluctuations term, $\Delta\epsilon_t$, as scaled white noise, that is,

$$\Delta\epsilon_t \propto \mathcal{N}(0, \mathbb{I})$$

The constant of proportionality was calculated so that the magnitude $\|\Delta\epsilon\|_2$ conformed to a pre-specified value. This magnitude was 2 in the simulation of *Figure 1*, and was a graphed variable in the simulations of *Figure 5* and *Figure 5—figure supplement 1*.

The compensatory plasticity term, $\Delta\mathbf{c}_t$, was calculated in two stages. First we applied the backpropagation algorithm, using $y(\mathbf{w}(0), u)$ as the ideal network outputs to train against. This generated an 'ideal' direction of compensatory plasticity , proportional to the negative gradient $\nabla F[\mathbf{w}(t)]$. For *Figure 5* and *Figure 5—figure supplement 1* we then corrupted this gradient with a tunable proportion of white noise. Overall, this gives,

$$\Delta\mathbf{c}_t = -\gamma_1\nabla\hat{F}[\mathbf{w}]_t + \%\gamma_2\hat{\nu}_t,$$

where $\nu_t \sim \mathcal{N}(0, \mathbb{I})$ is the noise corruption term, and $\gamma_1, \gamma_2 > 0$ are tunable hyperparameters. The higher the ratio $\gamma_2 : \gamma_1$, the greater the noise corruption. Meanwhile, $\sqrt{\gamma_1^2 + \gamma_2^2}$ sets the overall magnitude of compensatory plasticity . By tuning $\gamma_1$ and $\gamma_2$, we can therefore independently modify the magnitude and precision of the compensatory plasticity term. In *Figure 1*, we set $\gamma_2 = 0$.

## Acknowledgements

This work was supported by ERC grant StG 2016 716643 FLEXNEURO.

## Additional information

### Competing interests

Timothy O'Leary: Reviewing editor, *eLife*. The other author declares that no competing interests exist.

### Funding

| Funder | Grant reference number | Author |
| --- | --- | --- |
| European Commission | StG 2016 716643 FLEXNEURO | Dhruva V Raman Timothy O'Leary |

The funders had no role in study design, data collection and interpretation, or the decision to submit the work for publication.

### Author contributions

Dhruva V Raman, Conceptualization, Formal analysis, Investigation, Visualization, Methodology, Writing - original draft, Writing - review and editing; Timothy O'Leary, Conceptualization, Supervision,

Funding acquisition, Visualization, Writing - original draft, Project administration, Writing - review and editing

#### Author ORCIDs
Dhruva V Raman https://orcid.org/0000-0002-8992-1353
Timothy O'Leary https://orcid.org/0000-0002-1029-0158

#### Decision letter and Author response
Decision letter https://doi.org/10.7554/eLife.62912.sa1
Author response https://doi.org/10.7554/eLife.62912.sa2

## Additional files
### Supplementary files
• Transparent reporting form

### Data availability
All code is publicly available on github at this URL: https://github.com/Dhruva2/OptimalPlasticityRatios (copy archived at https://archive.softwareheritage.org/swh:1:rev:fcb1717a822f90b733c49d62bfc2f970155b7364).

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

## Appendix 1

### Alternative derivation of *Equation (9a)*

We provide an alternative derivation of *Equation (9a)* that removes the need for assumption (3b). We did not put this main derivation in the main text as we perceive it to have less clarity.

The derivation proceeds identically to that given in the main text until *Equation (7)*. We can then use (3a) to simplify *Equation (7)*. We get

$$\begin{aligned} \mathbb{E}[\Delta F] &= \Delta\mathbf{c}^T \nabla F[\mathbf{w}(t^*)] \\ &+ \frac{1}{2}\|\Delta\mathbf{c}\|_2^2 Q_{\mathbf{w}(t^*)}[\Delta\mathbf{c}] + \frac{1}{2}\|\Delta\epsilon\|_2^2 Q_{\mathbf{w}(t^*)}[\Delta\epsilon] \\ &+ \Delta\mathbf{c}^T(\nabla^2 F[\mathbf{w}(t^*)])\Delta\epsilon. \end{aligned}$$

$$\begin{aligned} \mathbb{E}[\Delta F] &= \Delta\mathbf{c}^T \nabla F[\mathbf{w}(t^*)] \\ &+ \frac{1}{2}\Delta\mathbf{c}^T(\nabla^2 F[\mathbf{w}(t^*)])\Delta\mathbf{c} + \frac{1}{2}\Delta\epsilon^T(\nabla^2 F[\mathbf{w}(t^*)])\Delta\epsilon \\ &+ \Delta\mathbf{c}^T(\nabla^2 F[\mathbf{w}(t^*)])\Delta\epsilon. \end{aligned}$$

Recall that expectation is taken over an unknown probability distribution from which $\Delta\epsilon$ is drawn, which satisfies *Equation (3a)*.

We then assume that we are in a phase of stable memory retention, so that $\mathbb{E}[\Delta F] = 0$. Now if the magnitude of compensatory plasticity $\|\Delta\mathbf{c}\|_2$ is tuned to minimise steady state error $F$, then any change to $\|\Delta\mathbf{c}\|_2$ will result in an increase in $\mathbb{E}[\Delta F]$. So $\mathbb{E}[\Delta F]$ is locally minimal in $\|\Delta\mathbf{c}\|_2$. This implies

$$\frac{\mathrm{d}\mathbb{E}[\Delta F]}{\mathrm{d}\|\Delta\mathbf{c}\|_2} = 0.$$

We also claim that local minimality implies

$$\frac{\mathrm{d}\mathbb{E}[\frac{\Delta F}{\|\Delta\mathbf{c}\|_2}]}{\mathrm{d}\|\Delta\mathbf{c}\|_2} = 0. \tag{1}$$

Why? $\mathbb{E}[\Delta F] = 0$ implies that $\mathbb{E}[\frac{\Delta F}{\|\Delta\mathbf{c}\|_2}] = 0$. If a small change to $\|\Delta\mathbf{c}\|_2$ results in $\mathbb{E}[\Delta F] \geq 0$, then it also results in $\mathbb{E}[\frac{\Delta F}{\|\Delta\mathbf{c}\|_2}] \geq 0$, since $\|\Delta\mathbf{c}\|_2$ is non-negative.

Expanding the LHS of *Equation (1)*, we get

$$\begin{aligned} \frac{\mathrm{d}}{\mathrm{d}\|\Delta\mathbf{c}\|_2} \Big\{ \Delta\hat{\mathbf{c}}^T \nabla F[\mathbf{w}(t^*)] &+ \frac{1}{2}\|\Delta\mathbf{c}\|_2 \Delta\hat{\mathbf{c}}^T(\nabla^2 F[\mathbf{w}(t^*)])\Delta\hat{\mathbf{c}} \\ &+ \frac{1}{2}\frac{\Delta\epsilon^T(\nabla^2 F[\mathbf{w}(t^*)])\Delta\epsilon}{\|\Delta\mathbf{c}\|_2} + \Delta\hat{\mathbf{c}}^T(\nabla^2 F[\mathbf{w}(t^*)])\Delta\epsilon \Big\} = 0. \end{aligned}$$

Differentiating, we get

$$\frac{1}{2}\Delta\hat{\mathbf{c}}^T(\nabla^2 F[\mathbf{w}(t^*)])\Delta\hat{\mathbf{c}} = \frac{1}{2}\frac{\Delta\epsilon^T(\nabla^2 F[\mathbf{w}(t^*)])\Delta\epsilon}{\|\Delta\mathbf{c}\|_2^2}.$$

$$\Rightarrow Q_{\mathbf{w}}[\Delta\mathbf{c}] = \frac{\|\Delta\epsilon\|_2^2}{\|\Delta\mathbf{c}\|_2^2} Q_{\mathbf{w}}[\Delta\epsilon],$$

from which (9a) follows.

### Positivity of the numerator and denominator in *Equation (9a)*

*Equation (9a)* of the main text asserts that

$$\frac{\|\Delta\mathbf{c}\|_2^2}{\|\Delta\epsilon\|_2^2} = \frac{Q_{\mathbf{w}}[\Delta\epsilon]}{Q_{\mathbf{w}}[\Delta\mathbf{c}]}$$

holds as long as both the numerator and denominator of the RHS are positive. Here we describe sufficient conditions for positivity.

The inequality $\nabla^2 F[\mathbf{w}] \succeq 0$ must hold in some neighbourhood of any minimum $\mathbf{w}^*$. Recall that we referred to such a neighbourhood as a highly trained state of the network in the main text. In such a state, our assertion follows immediately, as $Q_{\mathbf{w}}[\mathbf{v}] := \frac{1}{\|\mathbf{v}\|_2^2} \mathbf{v}^T (\nabla^2 F[\mathbf{w}]) \mathbf{v} \geq 0$, for any vector $\mathbf{v}$. Therefore, $Q_{\mathbf{w}}[\Delta\epsilon] \geq 0$ and $Q_{\mathbf{w}}[\Delta\mathbf{c}] \geq 0$.

We now consider a partially trained network state, which we defined in the main text as any $\mathbf{w}$ satisfying $Tr(\nabla^2 F) \geq 0$. Note that

$$F[\mathbf{w} + \Delta\epsilon] = F[\mathbf{w}] + \nabla F[\mathbf{w}]^T \Delta\epsilon + \frac{1}{2}\Delta\epsilon^T \nabla^2 F[\mathbf{w}]\Delta\epsilon + \mathcal{O}(\|\Delta\epsilon\|_2^3).$$

We assumed in the main text (*Equation (3a)*), that $\Delta\epsilon$ is uncorrelated with the gradient $\nabla F[\mathbf{w}]$ in expectation, since $\Delta\epsilon$ is realised by memory-independent processes. Similarly we can assume that $\Delta\epsilon$ is unbiased in how it projects onto the eigenvectors of $\nabla^2 F[\mathbf{w}]$. In other words,

$$\mathbb{E}[\hat{v}_i^T \Delta\epsilon] = \mathbb{E}[\hat{v}_j^T \Delta\epsilon],$$

for any normalised eigenvectors $\hat{v}_i$, $\hat{v}_j$ of $\nabla^2 F[\mathbf{w}]$. In expectation, we can therefore simplify to

$$\mathbb{E}[F[\mathbf{w} + \Delta\epsilon]] = F[\mathbf{w}] + \mathbb{E}[\nabla F[\mathbf{w}]^T \Delta\epsilon + \frac{1}{2}\Delta\epsilon^T \nabla^2 F[\mathbf{w}]\Delta\epsilon] + \mathcal{O}(\|\Delta\epsilon\|_2^3).$$

$$= 0 + \|\Delta\epsilon\|_2^2 \frac{Tr(\nabla^2 F[\mathbf{w}])}{N} + \mathcal{O}(\|\Delta\epsilon\|_2^3),$$

where $N$ is the dimensionality of the vector $\mathbf{w}$. So a partially trained network is one for which small, memory-independent weight fluctuations (such as $\Delta\epsilon$, or white noise) are expected to decrease task performance.

Now recall that $Q_{\mathbf{w}}[\Delta\epsilon] = \frac{1}{\|\Delta\epsilon\|_2^2}\Delta\epsilon^T \nabla^2 F[\mathbf{w}]\Delta\epsilon$. So we have

$$\mathbb{E}[Q_{\mathbf{w}}[\Delta\epsilon]] = \frac{Tr(\nabla^2 F[\mathbf{w}])}{N} > 0,$$

where the positivity constraint comes from being in a partially trained network.

We now consider why $Q_{\mathbf{w}}[\Delta\mathbf{c}]$ should be generically positive in a partially trained network. Suppose $Q_{\mathbf{w}}[\Delta\mathbf{c}] < 0$ holds. We can rewrite this as $\Delta\mathbf{c}^T \nabla^2 F[\mathbf{w}]\Delta\mathbf{c} \leq 0$. In this case, maintaining the same compensatory plasticity $\Delta\mathbf{c}$ over the time interval $[t^* + \Delta t, t^* + 2\Delta t]$ would result in increased improvement in loss, as

$$\nabla F[\mathbf{w} + \Delta\mathbf{c}]^T \Delta\mathbf{c} = \nabla F[\mathbf{w}]^T \Delta\mathbf{c} + \Delta\mathbf{c}^T \nabla^2 F[\mathbf{w}]\Delta\mathbf{c} + \mathcal{O}(\|\Delta\mathbf{c}\|_2^2).$$

Effectively, memory improvement due to compensatory plasticity $\Delta\mathbf{c}$ would be in an 'accelerating' direction, and maintaining the same direction $\Delta\mathbf{c}$ of compensatory plasticity would lead to ever faster learning. However, by assumption, we are in a regime of steady state task performance, where

$$\mathbb{E}[F(t^* + 2\Delta t) - F(t^* + \Delta t)] = \mathbb{E}[F(t^* + \Delta t) - F(t^*)] = 0.$$

## Optimal plasticity ratios in specific learning rules
### Noise-free learning rules (first-order)

Let us first consider the case where $\Delta\mathbf{c}$ can be computed with perfect access to the gradient $\nabla F[\mathbf{w}]$, but without access to $\nabla^2 F[\mathbf{w}]$. Such a $\Delta\mathbf{c}$ is known as a first-order learning rule, as it has access only to the first derivative of $F$ (*Polyak, 1987*). Imperfect access is considered subsequently. In this case, the optimal direction of compensatory plasticity is

$$\Delta\mathbf{c} \propto -\nabla F[\mathbf{w}].$$

In other words, $\Delta\mathbf{c}$ would implement perfect gradient descent on $F[\mathbf{w}]$. The condition of *Equation (11)* for synaptic fluctuations to outcompete reconsolidation plasticity evaluates to

$$Q_{\mathbf{w}}[\nabla F[\mathbf{w}]] \geq Q_{\mathbf{w}}[\Delta\epsilon].$$

To what extent can we quantify $Q_{\mathbf{w}}[\nabla F[\mathbf{w}]]$? First let us relate the gradient and Hessian of $F[\mathbf{w}]$. Let $\mathbf{w}^*$ be an optimal state of the network (i.e. one where $F$ is minimised). Let us parameterise the straight line connecting $\mathbf{w}$ with $\mathbf{w}^*$:

$$\gamma(s) = s\mathbf{w}^* + (1-s)\mathbf{w}, \quad s \in [0,1].$$

Then

$$\nabla F[\mathbf{w}] = (\mathbf{w} - \mathbf{w}^*)^T M, \text{ where}$$

$$M = \int_0^1 \nabla^2 F[\gamma(s)]\,ds.$$

This gives

$$Q_{\mathbf{w}}[\nabla F[\mathbf{w}]] = \frac{(\mathbf{w} - \mathbf{w}^*)^T M^T \nabla^2 F[\mathbf{w}] M (\mathbf{w} - \mathbf{w}^*)}{(\mathbf{w} - \mathbf{w}^*)^T M^T M (\mathbf{w} - \mathbf{w}^*)}.$$

First let us rewrite

$$(\mathbf{w} - \mathbf{w}^*) := \sum_i^N c_i v_i,$$

$$M(\mathbf{w} - \mathbf{w}^*) := \sum_i^N d_i v_i$$

where $(\lambda_i, v_i)$ is the $i^{th}$ eigenvalue/eigenvector pair of $\nabla^2 F$ (sorted in ascending order of $\lambda_i$), and $c_i$, $d_i$ are some scalar weights. Now

$$Q_{\mathbf{w}}[\nabla F[\mathbf{w}]] = \frac{\sum_{i=1}^N d_i^2 \lambda_i}{\sum_{i=1}^N d_i^2}. \tag{2}$$

The value of $Q_{\mathbf{w}}[\nabla F[\mathbf{w}]]$ now depends upon the distribution of mass of the sequence $\{d_i\}$. If later elements of the sequence are larger (i.e. $M(\mathbf{w} = \mathbf{w}^*)$ projects more highly onto eigenvectors of $\nabla^2 F[\mathbf{w}]$ with large eigenvalue), then $Q_{\mathbf{w}}[\nabla F[\mathbf{w}]]$ becomes larger, and the optimal magnitude of reconsolidation plasticity decreases, relative to the magnitude of synaptic fluctuations. The opposite is true if earlier elements of the sequence are larger.

Guaranteed bounds on the value of *Equation (2)* are vacuous. If we do not restrict $M$, then we can tailor the sequence $\{d_i\}$ as we like, and we end up with $\lambda_1 \leq Q_{\mathbf{w}}[\nabla F[\mathbf{w}]] \leq \lambda_N$. However, pragmatic bounds are much tighter. Let us now consider two plausibly extremal cases.

First consider the simplest case of a network that linearly transforms its outputs, and which has a quadratic loss function $F[\mathbf{w}]$. In this case $\nabla^2 F$ is a constant, (independent of $\mathbf{w}$), positive-semidefinite matrix, and $M = \nabla^2 F$. This means that

$$d_i = c_i \lambda_i v_i$$

$$Q_{\mathbf{w}}[\nabla F[\mathbf{w}]] = \frac{\sum_{i=1}^N c_i^2 \lambda_i^3}{\sum_{i=1}^N c_i^2 \lambda_i^2}.$$

Condition (11) then becomes

$$Q_{\mathbf{w}}[\nabla F[\mathbf{w}]] \geq Q_{\mathbf{w}}[\Delta\epsilon] \quad \Leftrightarrow \quad \frac{\sum_{i=1}^N c_i^2 \lambda_i^3}{\sum_{i=1}^N c_i^2 \lambda_i^2} \geq \frac{\sum_{i=1}^N \lambda_i}{N}. \tag{3}$$

A conservative sufficient condition for (*Equation 3*), using Chebyshev's summation inequality, is that

$$c_i^2 \lambda_i^2 \geq c_{i-1}^2 \lambda_{i-1}^2, \text{ for all } i \in \{1, \ldots, N\}. \tag{4}$$

Under what conditions would a plausible reconsolidation mechanism choose to 'outcompete' synaptic fluctuations, in this linear example? For $Q_\mathbf{w}[\nabla F[\mathbf{w}]] < Q_\mathbf{w}[\Delta\epsilon]$ to even hold, (**26**) would have to be broken, and significantly so due to conservatism in the inequality. In other words, $\mathbf{w} - \mathbf{w}^*$ must project quite biasedly onto the eigenvectors of $\nabla^2 F$ with smaller-than-average eigenvalue. If the discrepancy between $\mathbf{w}$ and $\mathbf{w}^*$ were caused by fluctuations (which are independent of $\nabla^2 F$), then this would not be the case, in expectation. Even if this were the case, the reconsolidation mechanism would have to know about the described bias. This requires knowledge of both $\mathbf{w}^*$ and $\nabla^2 F$, and is thus implausible.

Now let us consider the case of a generic nonlinear network. At one extreme, if $\|\mathbf{w} - \mathbf{w}^*\|_2$ is small, then $M \approx \nabla^2 F[\mathbf{w}]$, and the discussion of the linear case is valid. This corresponds to the case where steady state error is close to the minimum achievable by the network. As $\|\mathbf{w} - \mathbf{w}^*\|_2$ increases (i.e. steady state error gets worse), the correspondence between $M$ and $\nabla^2 F[\mathbf{w}]$ will likely decrease. Thus the optimal magnitude of reconsolidation plasticity, relative to the level of synaptic fluctuations, will rise.

We could consider another 'extreme' case in which $M$ and $\nabla^2 F[\mathbf{w}]$ were completely independent of each other. In this case,

$$d_i^2 \approx \frac{1}{N} \sum_{i=1}^{N} d_i^2. \tag{5}$$

In other words, the projection of $M(\mathbf{w} - \mathbf{w}^*)$ onto the different eigenvectors of $\nabla^2 F[\mathbf{w}]$ is approximately even. Using (24), this gives

$$Q_\mathbf{w}[\nabla F[\mathbf{w}]] \approx \frac{\sum_{i=1}^{N} \lambda_i}{N} = Q_\mathbf{w}[\Delta\epsilon].$$

In summary, we have two plausible extremes. One occurs where $M = \nabla^2 F[\mathbf{w}]$, and another occurs where $M$ is completely independent of $\nabla^2 F[\mathbf{w}]$. In either case, $Q_\mathbf{w}[\nabla F[\mathbf{w}]] \geq Q_\mathbf{w}[\Delta\epsilon]$, and so the magnitude of synaptic fluctuations should optimally outcompete/equal the magnitude of reconsolidation plasticity. Of course, there might be particular values of $\mathbf{w}$ where the correspondence between $M$ and $\nabla^2 F[\mathbf{w}]$ is 'worse' than chance. In other words, eigenvectors of $M$ with large eigenvalue preferentially project onto eigenvectors of $\nabla^2 F[\mathbf{w}]$ with small eigenvalue. In such cases, we would have $Q_\mathbf{w}[\nabla F[\mathbf{w}]] \leq Q_\mathbf{w}[\Delta\epsilon]$. However, we find it implausible that a reconsolidation mechanism would be able to gain sufficient information on $M$ to determine this at particular points in time, and thereby increase its plasticity magnitude.

## Noise-free learning rules (second-order)

Let us now suppose that $\Delta\mathbf{c}$ can be computed with perfect access to both $\nabla F[\mathbf{w}]$ and $\nabla^2 F[\mathbf{w}]$. In this case, the reconsolidation mechanism would optimally apply plasticity in the direction of the Newton step: we would have

$$\nabla^2 F[\mathbf{w}] \Delta\mathbf{c} = -\nabla F[\mathbf{w}].$$

Note that the Newton step is often conceptualised as a weighted form of gradient descent, where movement on the loss landscape is biased towards direction of lower curvature. Thus we would expect $Q_\mathbf{w}[\Delta\mathbf{c}]$ to be smaller, and the optimal proportion of reconsolidation plasticity to be larger. This is indeed the case. For mathematical tractability, we will restrict our discussion to the case in which $\nabla^2 F[\mathbf{w}] \succ 0$, and $M \succ 0$. This would hold if $F[\mathbf{w}]$ were convex, or if $\mathbf{w}$ were sufficiently close to a unique local minimum $\mathbf{w}^*$. In this case we can rewrite

$$\Delta \mathbf{c} = -\nabla^2 F[\mathbf{w}]^{-1} \nabla F[\mathbf{w}],$$

which gives

$$Q_{\mathbf{w}}[\Delta \mathbf{c}] = \frac{\nabla F[\mathbf{w}]^T (\nabla^2 F[\mathbf{w}])^{-1} \nabla F[\mathbf{w}]}{\nabla F[\mathbf{w}]^T (\nabla^2 F[\mathbf{w}])^{-2} \nabla F[\mathbf{w}]} \tag{6a}$$

$$= \frac{(\mathbf{w} - \mathbf{w}^*) M (\nabla^2 F[\mathbf{w}])^{-1} M (\mathbf{w} - \mathbf{w}^*)}{(\mathbf{w} - \mathbf{w}^*) M (\nabla^2 F[\mathbf{w}])^{-2} M (\mathbf{w} - \mathbf{w}^*)} \tag{6b}$$

$$= \frac{\sum_{i=1}^N d_i^2 \lambda_i^{-1}}{\sum_{i=1}^N d_i^2 \lambda_i^{-2}}. \tag{6c}$$

Once again, we first consider the case of a linear network with quadratic loss function, and hence with constant Hessian $\nabla^2 F$. This gives $M = \nabla^2 F$, and

$$Q_{\mathbf{w}}[\Delta \mathbf{c}] = \frac{(\mathbf{w} - \mathbf{w}^*) \nabla^2 F[\mathbf{w}] (\mathbf{w} - \mathbf{w}^*)}{\|\mathbf{w} - \mathbf{w}^*\|_2^2}$$

$$= \frac{\sum_{i=1}^N c_i^2 \lambda_i}{\sum_{i=1}^N c_i^2}.$$

We again assume that the reconsolidation mechanism does not have knowledge of the relative projections of $\mathbf{w} - \mathbf{w}^*$ onto the different eigenvectors of $\nabla^2 F$, which requires knowledge of $\mathbf{w}^*$. Without such information, we can use an analogous argument to that preceding (*Equation 5*) to argue that the approximation $c_i^2 \approx \frac{1}{N} \sum_{i=1}^N c_i^2$ is reasonable. This gives $Q_{\mathbf{w}}[\Delta \mathbf{c}] \approx Q_{\mathbf{w}}[\Delta \epsilon]$.

Note that the Newton step, in the linear-quadratic case just considered, corresponds to a direction $\mathbf{w}^* - \mathbf{w}$, that is, a direct path to a local minimum. So we could consider a compensatory plasticity mechanism implementing the Newton step as one directly undoing synaptic changes caused by $\Delta \epsilon$.

We now consider the case of a nonlinear network. As before, if $\|\mathbf{w} - \mathbf{w}^*\|_2$ is small, then we have $M \approx \nabla^2 F[\mathbf{w}]$, and the arguments of the linear network hold. As $\|\mathbf{w} - \mathbf{w}^*\|_2$ increases, the correspondence between $M$ and $\nabla^2 F$ will decrease. We again consider the plausible extreme where $M$ is completely uncorrelated with $\nabla^2 F[\mathbf{w}]$, and so the approximation (*Equation 5*) holds. In this case, *Equation (6)* can be simplified to give

$$Q_{\mathbf{w}}[\Delta \mathbf{c}] \approx \frac{\sum_{i=1}^N \lambda_i^{-1}}{\sum_{i=1}^N \lambda_i^{-2}}.$$

We assumed that $\nabla^2 F[\mathbf{w}] \succ 0$. Therefore, all eigenvalues are positive. This allows us to use Chebyshev's summation inequality to arrive at

$$\frac{\sum_{i=1}^N \lambda_i^{-1}}{\sum_{i=1}^N \lambda_i^{-2}} \leq \frac{\sum_{i=1}^N \lambda_i}{N} = Q_{\mathbf{w}}[\Delta \epsilon].$$

So as $\|\mathbf{w} - \mathbf{w}^*\|_2$ increases, the magnitude of reconsolidation plasticity will optimally outcompete that of synaptic fluctuations. This is the one case that contradicts our main claim.

## Imperfect learning rules

The previous section applied in the implausible case where a reconsolidation mechanism had perfect access to $\nabla F[\mathbf{w}]$ and/or $\nabla^2 F[\mathbf{w}]$. Recall from the main text that at least some information on $\nabla F[\mathbf{w}]$ is required, in order for compensatory plasticity to move in a direction of decreasing task error. What if $\Delta \mathbf{c}$ contains a mean-zero noise term, corresponding to unbiased noise corruption of these quantities? We will now show how such noise pushes $Q_{\mathbf{w}}[\Delta \mathbf{c}]$ towards equality with $Q_{\mathbf{w}}[\Delta \epsilon]$, and thus pushes the optimal magnitude of reconsolidation plasticity towards the magnitude of synaptic fluctuations. Let us use the model

$$\Delta \mathbf{c} = \tilde{\Delta \mathbf{c}} + \nu, \tag{7}$$

where $\nu$ is some mean-zero random variable, and $\tilde{\Delta \mathbf{c}}$ is the ideal output of the reconsolidation mechanism, assuming perfect access to the derivatives of $F[\mathbf{w}]$. Here $\nu$ represents the portion of compensatory plasticity attributable to systematic error in the algorithm, due to imperfect information on $F[\mathbf{w}]$. This could arise due to imperfect sensory information or limited communication between synapses. We can therefore assume, as for $\Delta \epsilon$, that it does not contain information on $\nabla^2 F[\mathbf{w}]$. We therefore get

$$Q_{\mathbf{w}}[\nu] \approx \frac{Tr(\nabla^2 F[\mathbf{w}])}{N},$$

analogously to *Equation (12)*. Now the operator $Q_{\mathbf{w}}$ satisfies

$$Q_{\mathbf{w}}[\Delta \mathbf{c}] = Q_{\mathbf{w}}[\tilde{\Delta \mathbf{c}}](1 + \frac{\|\nu\|_2^2}{\|\tilde{\Delta \mathbf{c}}\|_2^2})^{-1} + Q_{\mathbf{w}}[\nu](1 + \frac{\|\tilde{\Delta \mathbf{c}}\|_2^2}{\|\nu\|_2^2})^{-1}. \tag{8}$$

So depending upon the relative magnitudes of $\tilde{\Delta \mathbf{c}}$ and $\nu$, $Q_{\mathbf{w}}[\Delta \mathbf{c}]$ interpolates between $Q_{\mathbf{w}}[\tilde{\Delta \mathbf{c}}]$ and $Q_{\mathbf{w}}[\nu]$. In particular, as the crudeness of the learning rule (i.e. the ratio $\frac{\|\nu\|}{\|\Delta \mathbf{c}\|}$) grows, $Q_{\mathbf{w}}[\Delta \mathbf{c}]$ approaches equality (from below) with $Q_{\mathbf{w}}[\nu]$, and thus $Q_{\mathbf{w}}[\Delta \epsilon]$, completing our argument.

