## [Decision Letter]

**Acceptance summary:**

The half-century research on synaptic plasticity has primarily focused on how neural activity gives rise to changes in synaptic connections, and how these changes underlie learning and memory. However, recent studies have shown that in fact, most of the synaptic changes are activity-independent. This result is surprising given the generally held belief that activity-dependent changes in connectivity underlie network functionality. This manuscript proposes a theoretical explanation of why this should be the case. Specifically, this work presents a mathematical analysis of the amount of synaptic plasticity required to maintain learned circuit function in presence of random synaptic changes. The central finding, supported by simulations, is that for an "optimal" learning algorithm that rectifies random changes in connectivity, task-related plasticity should generally be smaller than the magnitude of the fluctuations. All reviewers agreed that this is a very interesting theoretical perspective on an important biological problem.

**Decision letter after peer review:**

Thank you for submitting your article "Optimal synaptic dynamics for memory maintenance in the presence of noise" for consideration by *eLife*. Your article has been reviewed by 3 peer reviewers, one of whom is a member of our Board of Reviewing Editors, and the evaluation has been overseen by Timothy Behrens as the Senior Editor. The following individuals involved in review of your submission have agreed to reveal their identity: Yonatan Loewenstein (Reviewer #2); Matthias H Hennig (Reviewer #3).

The reviewers have discussed the reviews with one another and the Reviewing Editor has drafted this decision to help you prepare a revised submission.

The half-century research on synaptic plasticity has primarily focused on how neural activity gives rise to changes in synaptic connections, and how these changes underlie learning and memory. However, recent studies have shown that in fact, most of the synaptic changes are activity-independent. This result is surprising given the generally held belief that activity-dependent changes in connectivity underlie network functionality. This manuscript proposes an explanation of why this should be the case.

Specifically, this work presents a mathematical analysis of the amount of synaptic plasticity required to maintain learned circuit function in presence of random synaptic changes. The central finding, supported by simulations, is that for an "optimal" learning algorithm that rectifies random changes in connectivity, task-related plasticity should generally be smaller than the magnitude of the fluctuations.

All reviewers agreed that this is a very interesting perspective on an important biological problem. However the reviewers have had difficulties fully understanding the specific claim, its derivation and potential implications. These issues are detailed in the Main Comments below, and need to be clarified. Additional suggestions are summarised in Other Comments.

Main comments:

1. How much does the main claim depend on the assumed learning algorithm? What is the family of learning algorithms that the authors refer to as "optimal" that have the property that directed changes are smaller in their magnitude than the noise-driven changes?

The reviewers' current understanding is the following, please clarify whether it is correct:

A) If the learning algorithm simply backtracks the noise, then the magnitude of learning-induced changes will be trivially equal that of the noise-induced changes. An optimal algorithm will not be worse than this trivial one.

B) If the learning algorithm (slowly) goes in the direction of the gradient of the objective function then (1) if the network is in a maximum of the objective function and changes are small then the magnitude of learning-induced changes will be equal to that of the noise-induced changes; (2) if the network is NOT in a maximum of the objective function then unless the noise is in the opposite direction to the gradient, the learning-induced path to the same level of performance would be shorter. Thus, the magnitude of learning-induced changes would be smaller that of that of the noise-induced changes

C) There exist (inefficient) learning algorithms such that the magnitude of the learning-induced changes are larger than the magnitude of the noise-induced ones. A learning algorithm that overshoots because of a too-large learning rate could be one of them, and it is trivial to construct other inefficient learning algorithms.

2. The reviewers have found the mathematical derivation in the main text difficult to follow, in part because it seems to use an unnecessarily complex set of mathematical notations and arguments. The reviewers suggest to focus on intuitive arguments in the main text, to make the arguments more transparent, and the paper more accessible to the broad neuroscience audience. Ultimately, this is up to authors to decide. In any case, the main arguments need to be clarified as laid out above.

3. The reviewers were not convinced by the role played by the numerical simulations. On one hand, it was not clear what the simulations added with respect to the analytical derivations. Is the purpose to check any specific approximation? On the other hand, the model does not seem to help link the derivations to the original biological question, as it is quite far removed from a biological setup. A minimal suggestion would be to use the model to illustrate more directly the geometric intuition behind the derivation, for instance by showing movements of weights along with some view of the gradient, contrasting optimal and sub-optimal.

Other comments:

4. The whole argument rests on an assumption that biological networks optimise a cost function, in particular during reconsolidation. How that assumption applies to the experimental setups detailed in the first part is unclear. At the very least, a clear statement and motivation of this assumption is needed.

5. One of the reviewers was not convinced (but happy to debate) cellular noise is such a major contributor to synaptic changes as stated in the introduction and in Table 1, as silencing neural activity pharmacologically will almost certainly affect synaptic function strongly. Strong synapses (big spines) tend to be more stable, and in any case it would be very difficult to know which of the observed modifications reported in the referenced papers have functional consequences. It would be very interesting to see what turnover (or weight dynamics) this model would predict under optimal and non-optimal conditions. In Figure 1 it is implied that weight changes continues unchanged in presence of noise (and optimal learning to maintain the objective), is this actually the case? What concrete experimental predictions the authors would (dare to) make?

[Editors' note: further revisions were suggested prior to acceptance, as described below.]

Thank you for resubmitting your work entitled "Optimal plasticity for memory maintenance in the presence of synaptic fluctuations" for further consideration by *eLife*. Your revised article has been evaluated by Timothy Behrens (Senior Editor) and a Reviewing Editor.

All reviewers have found that the manuscript has been very much improved and should be eventually published. The reviewers now fully understand the mathematical framework and the main mathematical result. During the consultation, it has however appeared that all reviewers feel that the main message of the manuscript, and in particular its implications for biology, need to be further clarified. Two main issues remain, which we suggest should be either addressed in the Results section or discussed in detail:

1. How much do the results rely on the assumption that synaptic changes are "strong"? To what extent is this assumption consistent with experiments? Is this theoretical framework really needed when infinitesimally small noise is immediately corrected by a learning signal, as often assumed (see below for more details)? Is the main result trivial when changes are small? Is the main contribution is to show that the result also holds far from this trivial regime, when noise and corrections are large?

2. what are the implications of the main mathematical result for interpreting measurable experimental quantities? The relation with experiments listed in the Discussion seems rather indirect (eg on lines 330-335 the interpretations of EM reconstructions in the authors' modelling framework seems unclear; it would be worth unpacking how the papers listed on lines 347-348 are consistent with the main result). Moreover, in many of the panels shown in Figures5-6, the dependence of the loss on the ratio between compensatory plasticity and synaptic fluctuations is rather flat; what does this imply for experimental data?

More details on the first point from one of the reviewers:

When studying learning in neuronal networks, the underlying assumption is always ("always" to the best of my knowledge) that learning-induced changes are gradual. For example, some form of activity-dependent plasticity has, on average, a negative projection on the gradient of the loss function of the current state. Small changes to synaptic efficacies are made and now the network is in a slightly different (improved) state. Activity-dependent plasticity in that new state has, again, a negative projection on the (new) gradient of the loss function at the new state, etc. If learning is sufficiently slow, we can average over the stochasticities in the learning process, organism's actions, rewards etc. and learning will improve performance.

In contrast to this approach, this paper suggests a very different learning process: activity-independent "noise" induces a LARGE change in the synaptic efficacies. This change is followed by a SINGLE LARGE compensatory learning-induced change. The question addressed in this manuscript is how large should this single optimal compensatory learning-induced change be relative to the single noise-induced change. The fact that the compensatory changes are not assumed to be small and that learning is done in a single step, rather than learning being gradual, allowing the local sampling of the loss function, complicates the mathematical analysis. While for analyzing infinitesimally-small changes we only need to consider the local gradient of the loss function, higher-order terms are required when considering single large changes.

What is the justification to this approach given that gradual learning is what we seem to observe in the experiments, specifically those cited in this manuscript? There is a lot of evidence of gradual changes in numbers of spines or synaptic efficacies, etc. If everything is gradual, why not "recompute" the gradient on the fly as is done in all previous models of learning?

---

## [Author Response]

Main comments:1. How much does the main claim depend on the assumed learning algorithm? What is the family of learning algorithms that the authors refer to as "optimal" that have the property that directed changes are smaller in their magnitude than the noise-driven changes?

We have made this much more clear in the main manuscript, with an additional table that summarises the families of learning algorithm satisfying the main claim.

Before proceeding, let us emphasise that we do not consider a learning algorithm itself as `optimal'. Learning algorithms induce both a direction and a magnitude of plasticity. We study the optimal magnitude of plasticity, for a given (probably imperfect) direction set by the learning algorithm. This leads to the equation:∥Δc∥2*2∥Δϵ∥22=Qw[Δϵ]Qw[Δc]in the main text, which is valid for any direction Δc of learning-induced plasticity.

Our main claim (learning-independent plasticity should outcompete learning-induced plasticity) follows as long as equation (1) is less than one. The value of equation (1), at steady-state error, does depend on the direction Δc of learning-induced plasticity and current network state (and hence on the learning algorithm).

For a network to exactly calculate equation (1), and thus work out the optimal magnitude of learning-induced plasticity, it would need to exactly calculate the Hessian ∇^2^F[w], which we consider implausible. Instead, the network could set the optimal magnitude using an expected value of (1) for an 'average' weight w. We calculate this value for different families of learning algorithm (see the new Table 2 in the main text). We now summarise these results (but they are also discussed in the section: Dependence of the optimal, steady-state magnitude of compensatory, learning-induced plasticity on learning algorithm).

The entire space of learning algorithms we consider is the space of incremental error based learning rules; that is, learning rules that induce small synaptic changes on small time intervals, for which the increment depends on some recent measure of error in a given task or deviation from some pre specified goal. This covers all standard rules assumed in theoretical studies: supervised, unsupervised and reinforcement learning that express weight change as a differential quantity.

We divide this space of learning algorithms into three cases: 0^th^-order algorithms (i.e. perturbation based), 1^st^-order algorithms (which approximate/calculate the gradient), and 2^nd^-order algorithms (which approximate/calculate both the gradient and the hessian).

In the case of a quadratic error function, we show that all three cases should should obey our main claim. We then note that nonlinear error functions, near a local minimum, should look quadratic, and thus have analogous conclusions. For nonlinear error functions far from a local minimum, we find that second order algorithms do not obey our main claim: learning-induced plasticity should exceed learning-independent plasticity at steady-state error. However, in the main text we question the biological plausibility of an accurate second order algorithm operating at a steady-state error far from a local minimum.

We have made the assumptions underlying the previously described results much more clear. In particular, we describe how we calculate our results for `perfect' (0^th^/1^st^/2^nd^)-order algorithms, and push these insights to `approximate' (0^th^/1^st^/2^nd^)-order algorithms by assuming that the approximation error term will not project onto the hessian ∇^2^F[w] more biasedly than the `perfect' component of the relevant term.

The reviewers' current understanding is the following, please clarify whether it is correct:A) If the learning algorithm simply backtracks the noise, then the magnitude of learning-induced changes will be trivially equal that of the noise-induced changes. An optimal algorithm will not be worse than this trivial one.

The first sentence is correct. The second sentence is not. The word `optimal' has been misconstrued: it applies to the magnitude of plasticity induced by a given learning algorithm, not the algorithm itself (which our results are essentially agnostic to, as explained above). Consider an arbitrary learning algorithm falling into the previously described space of algorithms we consider. The algorithm, especially if it obeys biological constraints, may be inefficient in selecting a `good' direction of plasticity (i.e. one that effectively decreases task error). Thus it may perform worse (in maintaining good steady-state error) than the ‘trivial' algorithm that backtracks the noise. It may also perform better, especially if the `backtracking' is onto some highly suboptimal network state. Regardless of how good the directions of plasticity induced by the learning algorithm is, there will be an optimal, associated magnitude of plasticity. We claim that this magnitude of plasticity should optimally be smaller or equal to the magnitude of ongoing synaptic fluctuations. We do not make claims about the direction of plasticity induced by different algorithms.

B) If the learning algorithm (slowly) goes in the direction of the gradient of the objective function then (1) if the network is in a maximum of the objective function and changes are small then the magnitude of learning-induced changes will be equal to that of the noise-induced changes; (2) if the network is NOT in a maximum of the objective function then unless the noise is in the opposite direction to the gradient, the learning-induced path to the same level of performance would be shorter. Thus, the magnitude of learning-induced changes would be smaller that of that of the noise-induced changes

This paragraph talks about gradient descent, which is one of the cases explored in the manuscript. Other cases are described in the main reply above. For the remainder of this reply, we assume that learning-induced plasticity is in the direction of the gradient of the objective function.

If the network is at steady-state task error, and close to a minimum w* of the task error (i.e. maximum of the objective function), we would expect the optimal magnitude of learning-induced plasticity to be less than that of the learning-independent plasticity. The more anisotropic the curvature (i.e. eigenvalues of ∇^2^F[w*]), the smaller this learning-induced magnitude should be, relative to the learning-independent plasticity. Extremally, it reaches equality with the magnitude of learning-induced plasticity when the eigenvalues of the latter matrix are completely isotropic.

The further the steady-state task error is from a minimum w*, the closer to parity we would expect the optimal magnitude of learning-induced plasticity to be, relative to the level of learning-independent plasticity.

We have revamped the figures, and in particular the new Figure 4 directly explains the geometric intuition behind this claim. Meanwhile, the section `Dependence of the optimal magnitude of steady-state, compensatory plasticity on the mechanism' justifies this claim.

C) There exist (inefficient) learning algorithms such that the magnitude of the learning-induced changes are larger than the magnitude of the noise-induced ones. A learning algorithm that overshoots because of a too-large learning rate could be one of them, and it is trivial to construct other inefficient learning algorithms.

Our understanding of the reviewer's comment is as follows: a learning algorithm could be `inefficient' for two reasons:

1. It selects noisy/imperfect directions of plasticity due to biological constraints. The degree to which this occurs in different learning systems is an open scientific question.

2. Given a (possibly imperfect) direction of plasticity, the accompanying magnitude of plasticity is too high/low. We agree that any learning algorithm could set the magnitude of plasticity associated with a particular direction too high. In this case the magnitude of learning-induced changes could indeed be larger than the learning-independent ones. By lowering the magnitude of learning-induced changes in this case, better steady-state task error would be achieved.

2. The reviewers have found the mathematical derivation in the main text difficult to follow, in part because it seems to use an unnecessarily complex set of mathematical notations and arguments. The reviewers suggest to focus on intuitive arguments in the main text, to make the arguments more transparent, and the paper more accessible to the broad neuroscience audience. Ultimately, this is up to authors to decide. In any case, the main arguments need to be clarified as laid out above.

We've tried out best to incorporate this suggestion. In particular, almost all of the maths is now contained in yellow boxes, which are separated from the main text. A reader who does not want to engage with the mathematics can read the entirety of the Results section without referring to the yellow boxes. We've expanded the description of the geometric intuition behind our results, and added new figures to help with this.

3. The reviewers were not convinced by the role played by the numerical simulations. On one hand, it was not clear what the simulations added with respect to the analytical derivations. Is the purpose to check any specific approximation? On the other hand, the model does not seem to help link the derivations to the original biological question, as it is quite far removed from a biological setup. A minimal suggestion would be to use the model to illustrate more directly the geometric intuition behind the derivation, for instance by showing movements of weights along with some view of the gradient, contrasting optimal and sub-optimal.

The numerical simulations themselves were there just to check the validity of the analytic derivations under different conditions (i.e. different magnitudes of learning-independent plasticity, and different accuracies of learning-induced plasticity). We agree that they did not provide much geometric intuition into the results, and that this geometric intuition was lacking in the original submission. We have rectified this by providing more detailed figures highlighting the geometric intuition (Figure 4 in particular). These depict your `minimal suggestion'. These were drawn, rather than derived by simulation, since they were depicting precise geometrical features of weight changes at a particular timepoint that we found difficult to cleanly show through simulation. They contrast `optimal' and `suboptimal', as requested, and provide intuition into why the optimal magnitude of learning-induced plasticity is usually lower than the fixed, learning-independent term. We added an extra `motivating' simulation in Figure 1.

The main claim of the paper is quite generic, and not specific to a particular circuit architecture and/or `biologically plausible' learning rule. We therefore decided to test the claim on an abstract, simple-to-explain setup, where we could easily and intuitively manipulate the `accuracy' of learning-induced plasticity. We decided not to run simulations on a more biologically-motivated circuit architecture/learning rule. If we had done so, we would have had to choose a particular, but arbitrary biologically-motivated setup. This would have required a detailed explanation that was unrelated to the point of the paper. It may have additionally confused casual readers as to the generic nature of the results. We have more clearly described the motivation behind the examples in the new section: Motivating example.

Other comments:4. The whole argument rests on an assumption that biological networks optimise a cost function, in particular during reconsolidation. How that assumption applies to the experimental setups detailed in the first part is unclear. At the very least, a clear statement and motivation of this assumption is needed.

We've rewritten this part of the paper to make this assumption much more explicit. We now list our exact assumptions at the beginning of the `Modelling setup' section. We note that any descriptive quantity such as `memory quality', or `learning performance', makes the implicit assumption of a loss function. Of course, the loss function may not be explicitly optimised by a neural circuit as the reviewer notes, but this does not actually matter from a mathematical point of view. As long as there is some organised state that a network evolves toward, one can posit an implicit loss function (or set of loss functions) that are being optimised in order to carry out the kind of analysis we performed here.

This is analogous to very widely known `cost functions' in physics: closed thermodynamic systems tend to maximise entropy over time; conservative mechanical systems minimise an abstract quantity called action. The components in these systems don't represent or interact with such abstract cost functions, yet the theories that assume them capture the relevant phenomena very successfully.

We would say that any view of reconsolidation that relies on retaining (to the greatest possible extent) some previously learned information, is implicitly trying to optimise a steady-state for some implicit loss function. Clearly, the notion of retaining previously learned information does not make sense in e.g. an in vitro setup. Nevertheless, the low-level plasticity mechanisms are likely to be have similarities to an intact setup, even if the signals they are receiving (e.g. patterns of incoming neural activity) are pathological. That said, we might speculate that synapses could tune the extent to which they respond to `endogenous' (task independent) signals versus external signals that could convey task information in the intact animal. If true, this is a potential way to interpret the in vitro data. We have included this point in the discussion.

We have reworded the introduction, making the link between experimental results and our analysis more clear. Note that we do not directly analyse the data provided by the relevant experimental papers. We explored the conclusions of a particular hypothesis: that neural circuits attempt to retain previously learned information in the face of synaptic fluctuations, through compensatory plasticity mechanisms. We then found that many experiments across the literature seemed consistent with our view, even if they don't conclusively imply it. Our work serves as a motivation to conduct further, specific experiments that attempt to isolate plasticity attributable to learning-independent mechanisms.

5. One of the reviewers was not convinced (but happy to debate) cellular noise is such a major contributor to synaptic changes as stated in the introduction and in Table 1, as silencing neural activity pharmacologically will almost certainly affect synaptic function strongly. Strong synapses (big spines) tend to be more stable, and in any case it would be very difficult to know which of the observed modifications reported in the referenced papers have functional consequences. It would be very interesting to see what turnover (or weight dynamics) this model would predict under optimal and non-optimal conditions. In Figure 1 it is implied that weight changes continues unchanged in presence of noise (and optimal learning to maintain the objective), is this actually the case? What concrete experimental predictions the authors would (dare to) make?

Firstly, we have reworded the manuscript to make more clear the fact that synaptic fluctuations may not only represent noise. They represent any plasticity process independent of the learned task: the probability of such a process increasing a weight is independent of whether such an increase is locally beneficial to task performance. Intrinsic processes such as homeostatic plasticity may also be important.

We agree that pharmacological silencing will strongly affect synaptic activity. Note that we also found experiments in the literature where the proportion of learning/activity-independent plasticity was estimated without pharmacological intervention. [2] looked at commonly innervated spines sharing a pre/post neuron, and by observing their in vitro weight dynamics found that the majority of such dynamics were accounted for by activity- independent processes. [2] also found the same conclusion in vivo by analysing commonly innervated synapses from an EM reconstruction of brain tissue from [3]. Meanwhile, [4] did a control experiment of raising mice in a visually impoverished environment, to compare against pharmacological silencing.

The literature does strongly suggest that large spines are more stable (e.g. [6]). We don't believe this contradicts any of the conclusions of the paper. We have added some more detail in the results and the discussion about how our results should be interpreted in the case where synaptic fluctuations are not distributed evenly within the neural circuit. In particular, the more that synaptic fluctuations are biased to more heavily alter less functionally important synapses, the lower the optimal magnitude of compensatory plasticity.

Note that even for simplified probabilistic models of evenly-distributed synaptic fluctuations, such as e.g. a white noise process, larger spines would still be more stable, as the probability of constant magnitude fluctuations eliminating a spine within a time period would decrease with spine size. In general, our results apply for any probabilistic form of synaptic fluctuation, as long as the probability of synaptic fluctuations increasing/decreasing a particular synaptic strength is independent of whether such a change would be beneficial for task performance.

Figure 1 one does imply that weight changes continue at steady state error. We have changed Figure 1 to show an explicit numerical simulation showing precisely that. As long as compensatory plasticity is not directly backtracking synaptic fluctuations, there will be an overall change in synaptic weights over time.

During learning, the magnitude of compensatory (i.e. learning) plasticity will be larger than at the point that steady state error is achieved. We have an extra section on the optimal magnitude of compensatory plasticity during learning describing this. Thus, the overall magnitude of plasticity (compensatory plus synaptic fluctuations) is predicted to be greater during learning.

A concrete experimental prediction follows from this observation that we now outline in the discussion: the proportion of learning-independent plasticity should be lower in a system that is actively learning, as opposed to retaining previously learned information. Interestingly, the literature seems to tentatively support this. Both [2] and [1] considered the covariance of functional synaptic strengths for co-innervated synapses, but in neocortex and hippocampus respectively. The hippocampal experiment showed much less activity- independent plasticity (compare Figure 1 of [1] to Figure 8 of [2]). This would make sense in light of our results if the hippocampus was in a phase of active learning, while the neocortex was in a phase of retaining previously learned information. In fact, many conventional cognitive theories of hippocampus and neocortex characterise the hippocampus as a continual, active learner, with the neocortex as a consolidator of previously learned information (see e.g. [5]).

References

[1] Thomas M Bartol Jr, Cailey Bromer, Justin Kinney, Michael A Chirillo, Jennifer N Bourne, Kristen M Harris, and Terrence J Sejnowski. Nanoconnectomic upper bound on the variability of synaptic plasticity. eLife, 4:e10778, 2015.

[2] Roman Dvorkin and Noam E. Ziv. Relative Contributions of Specific Activity Histories and Spontaneous Processes to Size Remodeling of Glutamatergic Synapses. PLOS Biology, 14(10):e1002572, 2016.

[3] Narayanan Kasthuri, Kenneth Jeffrey Hayworth, Daniel Raimund Berger, Richard Lee Schalek, Jose Angel Conchello, Seymour Knowles-Barley, Dongil Lee, Amelio Vazquez- Reina, Verena Kaynig, Thouis Raymond Jones, Mike Roberts, Josh Lyskowski Morgan, Juan Carlos Tapia, H. Sebastian Seung, William Gray Roncal, Joshua Tzvi Vogel- stein, Randal Burns, Daniel Lewis Sussman, Carey Eldin Priebe, Hanspeter Pfister, and Jeff William Lichtman. Saturated reconstruction of a volume of neocortex. Cell, 162(3):648{661, 2015.

[4] Akira Nagaoka, Hiroaki Takehara, Akiko Hayashi-Takagi, Jun Noguchi, Kazuhiko Ishii, Fukutoshi Shirai, Sho Yagishita, Takanori Akagi, Takanori Ichiki, and Haruo Kasai. Abnormal intrinsic dynamics of dendritic spines in a fragile X syndrome mouse model in vivo. Scientific Reports, 6:26651, 2016.

[5] Randall C. O'Reilly and Jerry W. Rudy. Conjunctive representations in learning and memory: principles of cortical and hippocampal function. Psychological review, 108(2):311, 2001.

[6] Nobuaki Yasumatsu, Masanori Matsuzaki, Takashi Miyazaki, Jun Noguchi, and Haruo Kasai. Principles of Long-Term Dynamics of Dendritic Spines. Journal of Neuroscience, 28(50):13592{13608, 2008.

[Editors' note: further revisions were suggested prior to acceptance, as described below.]

1. How much do the results rely on the assumption that synaptic changes are "strong"? To what extent is this assumption consistent with experiments? Is this theoretical framework really needed when infinitesimally small noise is immediately corrected by a learning signal, as often assumed (see below for more details)? Is the main result trivial when changes are small? Is the main contribution is to show that the result also holds far from this trivial regime, when noise and corrections are large?

There are a number of questions to unpack here. Before doing so we'd like to point out that the main results go beyond the calculations that corroborate magnitudes of fluctuations and systematic plasticity in experiments. The contributions include an analysis framework that lets us query and understand general relationships between learning rule quality and ongoing synaptic change without making detailed assumptions about circuit architecture and plasticity rules. For example, the relationships we derived reveal that fluctuations should dominate at steady state (significantly exceed 50% of total ongoing change) when a learning rule closely approximate the gradient of a loss function. Given the intense interest in whether approximations of gradient descent occur biologically in synaptic learning rules, this observation alone says that the high degree of turnover observed in some parts of the brain is in fact consistent with, and maybe regarded as circumstantial evidence for, gradient-like learning rules. We'd argue that this insight and the framework that provides it are far from trivial.

We now turn to the specifics of the reviewers' comment.

More details on the first point from one of the reviewers:When studying learning in neuronal networks, the underlying assumption is always ("always" to the best of my knowledge) that learning-induced changes are gradual. For example, some form of activity-dependent plasticity has, on average, a negative projection on the gradient of the loss function of the current state. Small changes to synaptic efficacies are made and now the network is in a slightly different (improved) state. Activity-dependent plasticity in that new state has, again, a negative projection on the (new) gradient of the loss function at the new state, etc.

It is an open question whether `gradual' change is the only way by which synaptic plasticity manifests experimentally. For instance, recent results from Jeff Magee's lab show that a single burst of synaptic plasticity over a single behavioural trial can activate place fields in mouse hippocampal CA1 neurons [1]. Moreover, this plasticity burst can be triggered even where there is a difference of seconds between the necessary factors of synaptic transmission and post-synaptic activation: the synapse integrates information over a long window before potentiating (or not). Even in more classical LTP papers from the last few decades that we now cite are somewhat equivocal on whether synaptic changes occur in one lump, or can be reduced to incremental changes. What is undeniable, empirically, is that a large change (e.g. 200% potentiation) can occur on a timescale of a few minutes, typical of most `induction windows'. Relative to behavioural timescales this is rather fast, so should it be modelled as gradual? In the end it might simply be mathematical convention or convenience that has led most theory papers to assume continuous changes. Fortunately, the setup of our paper accounts for both cases: continual, gradual change or temporally sparse bursts of synaptic plasticity. We have added detailed discussion points in the paper to make this clear.

If learning is sufficiently slow, we can average over the stochasticities in the learning process, organism's actions, rewards etc. and learning will improve performance.

We disagree with this statement in the context where learning-independent synaptic fluctuations are present. Learning must be fast enough to compensate for the synaptic fluctuations. If learning is arbitrarily slow, synaptic fluctuations will grow unchecked. How fast should learning (ie compensatory plasticity) be to have optimal learning performance in the presence of a given degree of synaptic fluctuations. That is the subject of the paper. We do agree with this statement in the context where the only source of stochasticity is the learning rule. In this case, the magnitude of stochasticity decreases with the speed of learning.

In contrast to this approach, this paper suggests a very different learning process: activity-independent "noise" induces a LARGE change in the synaptic efficacies. This change is followed by a SINGLE LARGE compensatory learning-induced change. The question addressed in this manuscript is how large should this single optimal compensatory learning-induced change be relative to the single noise-induced change. The fact that the compensatory changes are not assumed to be small and that learning is done in a single step, rather than learning being gradual, allowing the local sampling of the loss function, complicates the mathematical analysis. While for analyzing infinitesimally-small changes we only need to consider the local gradient of the loss function, higher-order terms are required when considering single large changes.

There is no sequential ordering of the compensatory plasticity and synaptic fluctuation terms, and we are not considering an alternating scenario, where a compensatory plasticity change reacts to a noisy change. Instead, we consider the relative proportions of the two, ongoing, plasticity terms over a (potentially infinitesimally small) time window. Our mathematics is in fact a first order, and not a second order analysis, and the results (in the regime of steady state task error) thus hold in the infinitesimal limit of small changes over the considered time window. We appreciate that this wasn't clear in the previous iteration, and have rewritten the Results section to make this more clear.

How can our analysis, which critically depends upon the Hessian (i.e. second order term) of the loss function, be a first order (i.e. linear) analysis, which is the relevant analysis in the limit of infinitesimally small changes? To find the optimal rate of compensatory plasticity, we have to differentiate the effect of compensatory plasticity with respect to its magnitude, and set the derivative equal to zero. This derivative-taking (unlabelled equation in Box 3, between equations 8 and 9) turns coefficients that were previously quadratic (second order) in the rate of compensatory plasticity (i.e. the Hessian), into first order (linear) coefficients. Indeed, our formula is locally linear: if we double the rate of fluctuations, it says that the optimal rate of compensatory plasticity should correspondingly double. The aforementioned derivative also turns the first order term in the loss function (i.e. the local gradient mentioned by the reviewer) into a zeroth order (constant) term, that is independent of plasticity rates. As a demonstration, suppose an ongoing compensatory plasticity mechanism (gradient descent, for simplicity) corrected the effects of white-noise, synaptic fluctuations. Meanwhile, task error was at a steady state F[w] = k. Over an infinitesimal time period δt, the white noise fluctuations changed the synaptic weights by a magnitude ϵ(δt). What rate of compensatory plasticity is required to cancel out the effect of white noise on the task error?

White noise is uncorrelated in expectation with the gradient of the task error. A first order Taylor expansion that only considers the local gradient would therefore give

E[F[w(*t*) + ϵ(δt)] - F[w(*t*)]] = E[∇*F*[w(*t*)]^T^ ϵ(*δt*)] = 0.

This analysis would suggest that the optimal rate of compensatory plasticity over δt is zero, since the synaptic fluctuations have no effect, to first order, on the task error. This is a zeroth order approximation of the optimal magnitude of compensatory plasticity. In other words, it is independent of the rate of synaptic fluctuations. They could double, and this analysis would still suggest that the optimal rate of compensatory plasticity should still be zero. This is why an analysis only considering the local gradient of the loss function is insufficient, even in the infinitesimal limit of small weight changes.

We have rewritten the Results section to make clear the correspondence between magnitudes of plasticity over small time intervals Δt, and instantaneous rates of plasticity, and have rephrased our terminology, where appropriate, to refer to plasticity `rates'.

What is the justification to this approach given that gradual learning is what we seem to observe in the experiments, specifically those cited in this manuscript? There is a lot of evidence of gradual changes in numbers of spines or synaptic efficacies, etc. If everything is gradual, why not "recompute" the gradient on the fly as is done in all previous models of learning?

In light of the previous comments, we can say that the manuscript is consistent with, and indeed describes, the gradual learning observed in the mentioned experiments. In the numerical simulations, each timestep corresponds to an “on the fly recomputation" of the (noisy) gradient, consistent with previous models of learning.

2. what are the implications of the main mathematical result for interpreting measurable experimental quantities? The relation with experiments listed in the Discussion seems rather indirect (eg on lines 330-335 the interpretations of EM reconstructions in the authors' modelling framework seems unclear; it would be worth unpacking how the papers listed on lines 347-348 are consistent with the main result). Moreover, in many of the panels shown in Figures5-6, the dependence of the loss on the ratio between compensatory plasticity and synaptic fluctuations is rather flat; what does this imply for experimental data?

We have expanded the discussion on how our modelling framework relates to the mentioned papers. In particular, we have spelled out how our notion of `compensatory plasticity' may be approximated using

– the covariance in synaptic strengths for co-innervated synapses in EM reconstructions;

– the ‘activity-independent' plasticity in experiments that suppress neural activity,

We also go into greater detail about the biological assumptions inherent in our modelling. Even more detail is provided in the first subsection of the Results (review of key experimental findings), as we are aware of the need to keep the discussion reasonably concise.

As to the relative flatness of the curves in Figure 6, this is for a linear network and is included for mathematical completeness. We have now made this a supplement to Figure 5 (nonlinear network) which is likely more relevant biologically. In Figure 5 itself, the reviewers will note that the relationship is only at for an interval of relatively poor quality learning rules (middle row, where steady state error is almost 2 orders of magnitude worse than the case in the top row, which itself has a learning rule with a correlation of only 0.1 with a gradient). We included this along with the third row (where the dependence is once again steep) to show the general trend, in which a strong U-shape is more typical. In any case, a flat dependence for high compensatory plasticity, while consistent with the theory in certain regimes, is less consistent with the experimental data we reviewed and sought to account for.

Additional commentLines 180, 198: Figure 3d is referred to from the text but seems to be missing!

We corrected this typo, which occurred when we merged panels c and d in a draft copy of Figure 3.

References

[1] Katie C Bittner, Aaron D Milstein, Christine Grienberger, Sandro Romani, and Jeffrey C Magee. Behavioral time scale synaptic plasticity underlies ca1 place fields. *Science*, 357(6355):1033{1036, 2017.